# Learnability of Influence in Networks

**Harikrishna Narasimhan**     **David C. Parkes**     **Yaron Singer**
Harvard University, Cambridge, MA 02138
hnarasimhan@seas.harvard.edu, {parkes, yaron}@seas.harvard.edu

## Abstract

We show PAC learnability of influence functions for three common influence models, namely, the Linear Threshold (LT), Independent Cascade (IC) and Voter models, and present concrete sample complexity results in each case. Our results for the LT model are based on interesting connections with neural networks; those for the IC model are based an interpretation of the influence function as an expectation over random draw of a subgraph and use covering number arguments; and those for the Voter model are based on a reduction to linear regression. We show these results for the case in which the cascades are only partially observed and we do not see the time steps in which a node has been influenced. We also provide efficient polynomial time learning algorithms for a setting with full observation, i.e. where the cascades also contain the time steps in which nodes are influenced.

## 1   Introduction

For several decades there has been much interest in understanding the manner in which ideas, language, and information cascades spread through society. With the advent of social networking technologies in recent years, digital traces of human interactions are becoming available, and the problem of predicting information cascades from these traces has gained enormous practical value. For example, this is critical in applications like viral marketing, where one needs to maximize awareness about a product by selecting a small set of influential users [1].

To this end, the spread of information in networks is modeled as an *influence function* which maps a set of seed nodes who initiate the cascade to (a distribution on) the set of individuals who will be influenced as a result [2]. These models are parametrized by variables that are unknown and need to be estimated from data. There has been much work on estimating the parameters of influence models (or the structure of the underlying social graph) from observed cascades of influence spread, and on using the estimated parameters to predict influence for a given seed set [3, 4, 5, 6, 7, 8]. These parameter estimation techniques make use of local influence information at each node, and there has been a recent line of work devoted to providing sample complexity guarantees for these local estimation techniques [9, 10, 11, 12, 13].

However, one cannot locally estimate the influence parameters when the cascades are not completely observed (e.g. when the cascades do not contain the time at which the nodes are influenced). Moreover, influence functions can be sensitive to errors in model parameters, and existing results do not tell us to what accuracy the individual parameters need to be estimated to obtain accurate influence predictions. If the primary goal in an application is to predict influence accurately, it is natural to ask for algorithms that have learnability guarantees on the influence function itself. A benchmark for studying such questions is the Probably Approximately Correct (PAC) learning framework [14]:

*Are influence functions PAC learnable?*

While many influence models have been popularized due to their approximation guarantees for influence maximization [2, 15, 16], learnability of influence is an equally fundamental property.

---

Part of this work was done when HN was a PhD student at the Indian Institute of Science, Bangalore.

In this paper, we show PAC learnability for three well-studied influence models: the Linear Threshold, the Independent Cascade, and the Voter models. We primarily consider a setting where the cascades are *partially* observed, i.e. where only the nodes influenced and not the time steps at which they were influenced are observed. This is a setting where existing local estimation techniques cannot be applied to obtain parameter estimates. Additionally, for a *fully* observed setting where the time of influence is also observed, we show polynomial time learnability; our methods here are akin to using local estimation techniques, but come with guarantees on the global influence function.

**Main results.** Our learnability results are summarized below.

- **Linear threshold (LT) model:** Our result here is based on an interesting observation that LT influence functions can be seen as multi-layer neural network classifiers, and proceed by bounding their VC-dimension. The method analyzed here picks a function with zero training error. While this can be computationally hard to implement under partial observation, we provide a polynomial time algorithm for the full observation case using local computations.

- **Independent cascade (IC) model:** Our result uses an interpretation of the influence function as an expectation over random draw of a subgraph [2]; this allows us to show that the function is Lipschitz and invoke covering number arguments. The algorithm analyzed for partial observation is based on global maximum likelihood estimation. Under full observation (and additional assumptions), we show polynomial time learnability using a local estimation technique.

- **Voter model:** Our result follows from a reduction of the learning problem to a linear regression problem; the resulting learning algorithm can be implemented in polynomial time for both the full and partial observation settings.

**Related work.** A related problem to ours is that of inferring the structure of the underlying social graph from cascades [6]. There has been a series of results on polynomial sample complexity guarantees for this problem under variants of the IC model [9, 12, 10, 11]. Most of these results make specific assumptions on the cascades/graph structure, and assume a full observation setting. On the other hand, in our problem, the structure of the social graph is assumed to be known, and the goal is to provably learn the underlying influence function. Our results do not depend on assumptions on the network structure, and primarily apply to the more challenging partial observation setting.

The work that is most related to ours is that of Du et al. [13], who show polynomial sample complexity results for learning influence in the LT and IC models (under partial observation). However, their approach uses approximations to influence functions and consequently requires a strong technical condition to hold, which is not necessarily satisfied in general. Our results for the LT and IC models are some what orthogonal. While the authors in [13] trade-off assumptions on learnability and gain efficient algorithms that work well in practice, our goal is to show unconditional sample complexity for learning influence. We do this at the expense of the efficiency of the learning algorithms in the partial observation setting. Moreover, the technical approach we take is substantially different.

There has also been work on learnability of families of discrete functions such as submodular [17] and coverage functions [18], under the PAC and the variant PMAC frameworks. These results assume availability of a training sample containing *exact* values of the target function on the given input sets. While IC influence functions can be seen as coverage functions [2], the previous results do not directly apply to the IC class, as in practice, the true (expected) value of an IC influence function on a seed set is never observed, and only a *random* realization is seen. In contrast, our learnability result for IC functions do not require the exact function values to be known. Moreover, the previous results require strict assumptions on the input distribution. Since we focus on learnability of specific function classes rather than large families of discrete functions, we are able to handle general seed distributions for most part. Other results relevant to our work include learnability of linear influence games [19], where the techniques used bear some similarity to our analysis for the LT model.

## 2 Preliminaries

**Influence models.** We represent a social network as a finite graph $G = (V, E)$, where the nodes $V = \{1, \ldots, n\}$ represent a set of $n$ individuals and edges $E \subseteq V^2$ represent their social links. Let $|E| = r$. The graph is assumed to be directed unless otherwise specified. Each edge $(u, v) \in E$ is associated with a weight $w_{uv} \in \mathbb{R}_+$ that indicates the strength of influence of node $v$ on node $u$. We consider a setting where each node in the network holds an opinion in $\{0, 1\}$ and opinions

disseminate in the network. This dissemination process begins with a small subset of nodes called the *seed* which have opinion 1 while the rest have opinion 0, and continues in discrete time steps. In every time step, a node may change its opinion from 0 to 1 based on the opinion of its neighbors, and according to some *local* model of influence; if this happens, we say that the node is *influenced*. We will use $N(u)$ to denote the set of neighbors of node $u$, and $A_t$ to denote the set of nodes that are influenced at time step $t$. We consider three well-studied models:

- **Linear threshold (LT) model:** Each node $u$ holds a *threshold* $r_u \in \mathbb{R}_+$, and is influenced at time $t$ if the total incoming weight from its neighbors that were influenced at the previous time step $t-1$ exceeds the threshold: $\sum_{v \in N(u) \cap A_{t-1}} w_{uv} \geq k_u$. Once influenced, node $u$ can then influence its neighbors for one time step, and never changes its opinion to 0.[1]

- **Independent cascade (IC) model:** Restricting edge weights $w_{uv}$ to be in $[0,1]$, a node $u$ is influenced at time $t$ independently by each neighbor $v$ who was influenced at time $t-1$. The node can then influence its neighbors for one time step, and never changes its opinion to 0.

- **Voter model:** The graph is assumed to be undirected (with self-loops); at time step $t$, a node $u$ adopts the opinion of its neighbor $v$ with probability $w_{uv}/\sum_{v' \in N(u) \cup \{u\}} w_{uv'}$. Unlike the LT and IC models, here a node may change its opinion from 1 to 0 or 0 to 1 at every step.

We stress that a node is influenced **at** time $t$ if it changes its opinion from 0 to 1 exactly at $t$. Also, in both the LT and IC models, *no node gets influenced more than once and hence an influence cascade can last for at most $n$ time steps*. For simplicity, we shall consider in all our definitions only cascades of length $n$. While revisiting the Voter model in Section 5, we will look at more general cascades.

**Definition 1** (**Influence function**). *Given an influence model, a (global) influence function $F$: $2^V \to [0,1]^n$ maps an initial set of nodes $X \subseteq V$ seeded with opinion 1 to a vector of probabilities $[F_1(X), \ldots, F_n(X)] \in [0,1]^n$, where the $u^{th}$ coordinate indicates the probability of node $u \in V$ being influenced during **any time step** of the corresponding influence cascades.*

Note that for the LT model, the influence process is deterministic, and the influence function simply outputs a binary vector in $\{0,1\}^n$. Let $\mathcal{F}_G$ denote the class of all influence functions under an influence model over $G$, obtained for different choices of parameters (edge weights/thresholds) in the model. We will be interested in learning the influence function for a given parametrization of this influence model. We shall assume that the initial set of nodes that are seeded with opinion 1 at the start of the influence process, or the seed set, is chosen i.i.d. according to a distribution $\mu$ over all subsets of nodes. We are given a training sample consisting of draws of initial seed sets from $\mu$, along with observations of nodes influenced in the corresponding influence process. Our goal is to then learn from $\mathcal{F}_G$ an influence function that best captures the observed influence process.

**Measuring Loss.** To measure quality of the learned influence function, we define a *loss function* $\ell : 2^V \times [0,1]^n \to \mathbb{R}_+$ that for any subset of influenced nodes $Y \subseteq V$ and predicted influence probabilities $\mathbf{p} \in [0,1]^n$ assigns a value $\ell(Y, \mathbf{p})$ measuring discrepancy between $Y$ and $\mathbf{p}$. We define the error of a learned function $F \in \mathcal{F}_G$ for a given seed distribution $\mu$ and model parametrization as the expected loss incurred by $F$:

$$\text{err}^\ell[F] = \mathbf{E}_{X,Y}\big[\ell\big(Y, F(X)\big)\big],$$

where the above expectation is over a random draw of the seed set $X$ from distribution $\mu$ and over the corresponding subsets of nodes $Y$ influenced during the cascade.

We will be particularly interested in the difference between the error of an influence function $F_S \in \mathcal{F}_G$ learned from a training sample $S$ and the minimum possible error achievable over all influence functions in $\mathcal{F}_G$: $\text{err}^\ell\big[F_S\big] - \inf_{F \in \mathcal{F}_G} \text{err}^\ell\big[F\big]$, and would like to learn influence functions for which this difference is guaranteed to be small (using only polynomially many training examples).

**Full and partial observation.** We primarily work in a setting in which we observe the nodes influenced in a cascade, but not the time step at which they were influenced. In other words, we assume availability of a *partial observed* training sample $S = \{(X^1, Y^1) \ldots, (X^m, Y^m)\}$, where $X^i$ denotes the seed set of a cascade $i$ and $Y^i$ is the set of nodes influenced in that cascade. We will also consider a refined notion of *full observation* in which we are provided a training sample $S = \{(X^1, Y_{1:n}^1) \ldots, (X^m, Y_{1:n}^m)\}$, where $Y_{1:n}^i = \{Y_1^i, \ldots, Y_n^i\}$ and $Y_t^i$ is the set of nodes in

cascade $i$ who were influenced precisely at time step $t$. Notice that here the complete set of nodes influenced in cascade $i$ is given by $\bigcup_{t=1}^{n} Y_t^i$. This setting is particularly of interest when discussing learnability in polynomial time. The structure of the social graph is always assumed to be known.

**PAC learnability of influence functions.** Let $\mathcal{F}_G$ be the class of all influence functions under an influence model over a $n$-node social network $G = (V, E)$. We say $\mathcal{F}_G$ is *probably approximately correct (PAC) learnable w.r.t. loss $\ell$* if there exists an algorithm s.t. the following holds for $\forall \epsilon, \delta \in (0, 1)$, for all parametrizations of the model, and for all (or a subset of) distributions $\mu$ over seed sets: when the algorithm is given a partially observed training sample $S = \{(X^1, Y^1), \dots, (X^m, Y^m)\}$ with $m \geq \text{poly}(1/\epsilon, 1/\delta)$ examples, it outputs an influence function $F_S \in \mathcal{F}_G$ for which

$$\mathbf{P}_S\Big(\text{err}^\ell[F_S] - \inf_{F \in \mathcal{F}_G} \text{err}^\ell[F] \geq \epsilon\Big) \leq \delta,$$

where the above probability is over the randomness in $S$. Moreover, $\mathcal{F}_G$ is *efficiently PAC learnable* under this setting if the running time of the algorithm in the above definition is polynomial in $m$ and in the size of $G$. We say $\mathcal{F}_G$ is *(efficiently) PAC learnable under full observation* if the above definition holds with a fully observed training sample $S = \{(X^1, Y_{1:n}^1), \dots, (X^m, Y_{1:n}^m)\}$.

**Sensitivity of influence functions to parameter errors.** A common approach to predicting influence under full observation is to estimate the model parameters using local influence information at each node. However, an influence function can be highly sensitive to errors in estimated parameters. E.g. consider an IC model on a chain of $n$ nodes where all edge parameters are 1; if the parameters have all been underestimated with a constant error of $\epsilon$, the estimated probability of the last node being influenced is $(1-\epsilon)^n$, which is exponentially smaller than the true value 1 for large $n$. Our results for full observation provide concrete sample complexity guarantees for learning influence functions using local estimation, to any desired accuracy; in particular, for the above example, our results prescribe that $\epsilon$ be driven below $1/n$ for accurate predictions (see Section 4 on IC model). Of course, under partial observation, we do not see enough information to locally estimate the individual model parameters, and the influence function needs to be learned directly from cascades.

## 3    The Linear Threshold model

We start with learnability in the Linear Threshold (LT) model. Given that the influence process is deterministic and the influence function outputs binary values, we use the 0-1 loss for evaluation; for any subset of nodes $Y \subseteq V$ and predicted boolean vector $\mathbf{q} \in \{0, 1\}^n$, this is the fraction of nodes on which the prediction is wrong: $\ell_{0\text{-}1}(Y, \mathbf{q}) = \frac{1}{n} \sum_{u=1}^{n} \mathbf{1}(\chi_u(Y) \neq q_u)$, where $\chi_u(Y) = \mathbf{1}(u \in Y)$.

**Theorem 1** (**PAC learnability under LT model**). *The class of influence functions under the LT model is PAC learnable w.r.t. $\ell_{0\text{-}1}$ and the corresponding sample complexity is $\widetilde{O}\big(\epsilon^{-1}(r+n)\big)$. Furthermore, in the full observation setting the influence functions can be learned in polynomial time.*

The proof is in Appendix A and we give an outline here. Let $F^{\mathbf{w}}$ denote a LT influence function with parameters $\mathbf{w} \in \mathbb{R}^{r+n}$ (edge weights and thresholds) and let us focus on the partial observation setting (only a node and not its time of influence is observed). Consider a simple algorithm that outputs an influence function with zero error on training sample $S = \{(X^1, Y^1), \dots, (X^m, Y^m)\}$:

$$\frac{1}{m} \sum_{i=1}^{m} \ell_{0\text{-}1}\big(Y^i, F^{\mathbf{w}}(X^i)\big) = \frac{1}{mn} \sum_{i=1}^{m} \sum_{u=1}^{n} \mathbf{1}\big(\chi_u(Y^i) \neq F_u^{\mathbf{w}}(X^i)\big). \tag{1}$$

Such a function always exists as the training cascades are generated using the LT model. We will shortly look at computational issues in implementing this algorithm. We now explain our PAC learnability result for this algorithm. The main idea is in interpreting LT influence functions as neural networks with linear threshold activations. The proof follows by bounding the VC-dimension of the class of all functions $F_u^{\mathbf{w}}$ for node $u$, and using standard arguments in showing learnability under finite VC-dimension [20]. We sketch the neural network (NN) construction in two steps (local influence as a two-layer NN, and the global influence as a multilayer network; see Figure 1), where a crucial part is in ensuring that no node gets influenced more than once during the influence process:

1. **Local influence as a two-layer NN.** Recall that the (local) influence at a node $u$ for previously influenced nodes $Z$ is given by $\mathbf{1}\big(\sum_{v \in N(u) \cap Z} w_{uv} \geq k_u\big)$. This can be modeled as a linear (binary) classifier, or equivalently as a two-layer NN with linear threshold activations. Here the input layer contains a unit for each node in the network and takes a binary value indicating whether the node

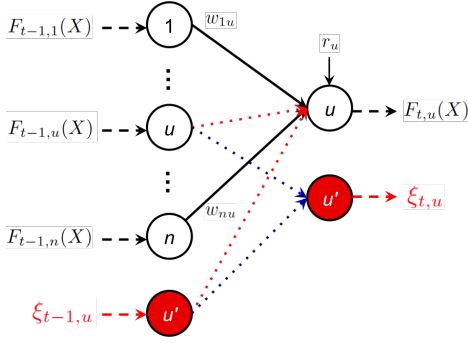

Figure 1: Modeling a single time step $t$ of the influence process $F_{t,u} : 2^V \to \{0, 1\}$ as a neural network ($t \geq 2$): the portion in black computes whether or not node $u$ is influenced in the current time step $t$, while that in red/blue enforces the constraint that $u$ does not get influenced more than once during the influence process. Here $\xi_{t,u}$ is 1 when a node has been influenced previously and 0 otherwise. The dotted red edges represent strong negative signals (has a large negative weight) and the dotted blue edges represent strong positive signals. The initial input to each node $u$ in the input layer is $\mathbf{1}(u \in X)$, while that for the auxiliary nodes (in red) is 0.

is present in $Z$; the output layer contains a binary unit indicating whether $u$ is influenced after one time step; the connections between the two layers correspond to the edges between $u$ and other nodes; and the threshold term on the output unit is the threshold parameter $k_u$. Thus the first step of the influence process can be modeled using a NN with two $n$-node layers (the input layer takes information about the seed set, and the binary output indicates which nodes got influenced).

2. **From local to global: the multilayer network.** The two-layer NN can be extended to multiple time steps by replicating the output layer once for each step. However, the resulting NN will allow a node to get influenced more than once during the influence process. To avoid this, we introduce an additional binary unit $u'$ for each node $u$ in a layer, which will record whether node $u$ was influenced in previous time steps. In particular, whenever node $u$ is influenced in a layer, a strong positive signal is sent to activate $u'$ in the next layer, which in turn will send out strong negative signals to ensure $u$ is never activated in subsequent layers[2]; we use additional connections to ensure that $u'$ remains active there after. Note that a node $u$ in layer $t + 1$ is 1 whenever $u$ is influenced at time step $t$; let $F_{t,u}^{\mathbf{w}} : 2^V \to \{0, 1\}$ denote this function computed at $u$ for a given seed set. The LT influence function $F_u^{\mathbf{w}}$ (which for seed set $X$ is 1 whenever $u$ is influenced in any one of the $n$ time steps) is then given by $F_u^{\mathbf{w}}(X) = \sum_{t=1}^n F_{t,u}^{\mathbf{w}}(X)$. Clearly, $F_u^{\mathbf{w}}$ can be modeled as a NN with $n + 1$ layers.

A naive application of classic VC-dimension results for NN [21] will give us that the VC-dimension of the class of functions $F_u$ is $\widetilde{O}(n(r + n))$ (counting $r + n$ parameters for each layer). Since the same parameters are repeated across layers, this can be tightened to $\widetilde{O}(r + n)$. The remaining proof involves standard uniform convergence arguments [20] and a union bound over all nodes.

## 3.1 Efficient computation

Having shown PAC learnability, we turn to efficient implementation of the prescribed algorithm.

**Partial observation.** In the case where the training set does not specify the time at which each node was infected, finding an influence function with zero training error is computationally hard in general (as this is similar to learning a recurrent neural network). In practice, however, we can leverage the neural network construction, and solve the problem *approximately* by replacing linear threshold activation functions with sigmoidal activations and the 0-1 loss with a suitable continuous surrogate loss, and apply back-propagation based methods used for neural network learning.

**Full observation.** Here it turns out that the algorithm can be implemented in polynomial time using *local* computations. Given a fully observed sample $S = \{(X^1, Y_{1:n}^1), \ldots, (X^m, Y_{1:n}^m)\}$, the loss of an influence function $F$ for any $(X, Y_{1:n})$ is given by $\ell_{0\text{-}1}(\cup_{t=1}^n Y_t, F(X))$ and as before measures the fraction of mispredicted nodes. The prescribed algorithm then seeks to find parameters $\mathbf{w}$ for which the corresponding training error is 0. Given that the time of influence is observed, this problem can be decoupled into a set of linear programs (LPs) at each node; this is akin to locally estimating the parameters at each node. In particular, let $\mathbf{w}_u$ denote the parameters local to node $u$ (incoming weights and threshold), and let $f_u(Z; \mathbf{w}_u) = \mathbf{1}\big(\sum_{v \in N(u) \cap Z} w_{uv} \geq k_u\big)$ denote the local influence at $u$ for set $Z$ of previously influence nodes. Let $\widehat{\alpha}_{1,u}(\mathbf{w}_u) = \frac{1}{m} \sum_{i=1}^m \mathbf{1}\big(\chi_u(Y_1^i) \neq f_u(X^i; \mathbf{w}_u)\big)$ and $\widehat{\alpha}_{t,u}(\mathbf{w}_u) = \frac{1}{m} \sum_{i=1}^m \mathbf{1}\big(\chi_u(Y_t^i) \neq f_u(Y_{t-1}^i; \mathbf{w}_u)\big)$, $t \geq 2$, that given the set of nodes $Y_{t-1}^i$ influenced at time $t - 1$, measures the local prediction error at time $t$. Since the training sample was

generated by a LT model, there always exists parameters such that $\widehat{\alpha}_{t,u}(\mathbf{w}_u) = 0$ for each $t$ and $u$, which also implies that the overall training error is 0. Such a set of parameters can be obtained by formulating a suitable LP that can be solved in polynomial time. The details are in Appendix A.2.

## 4 The Independent Cascade model

We now address the question of learnability in the Independent Cascade (IC) model. Since the influence functions here have probabilistic outputs, the proof techniques we shall use will be different from the previous section, and will rely on arguments based on covering numbers. In this case, we use the squared loss which for any $Y \subseteq V$ and $\mathbf{q} \in [0,1]^n$, is given by: $\ell_{\mathrm{sq}}(Y, \mathbf{q}) = \frac{1}{n} \sum_{u=1}^{n} [\chi_u(Y)(1 - q_u)^2 + (1 - \chi_u(Y))q_u^2]$. We shall make a mild assumption that the edge probabilities are bounded away from 0 and 1, i.e. $\mathbf{w} \in [\lambda, 1 - \lambda]^r$ for some $\lambda \in (0, 0.5)$.

**Theorem 2** (**PAC learnability under IC model**)**.** *The class of influence functions under the IC model is PAC learnable w.r.t. $\ell_{\mathrm{sq}}$ and the sample complexity is $m = \widetilde{O}(\epsilon^{-2} n^3 r)$. Furthermore, in the full observation setting, under additional assumptions (see Assumption 1), the influence functions can be learned in polynomial time with sample complexity $\widetilde{O}(\epsilon^{-2} n r^3)$.*

The proof is given in Appendix B. As noted earlier, an IC influence function can be sensitive to errors in estimated parameters. Hence before discussing our algorithms and analysis, we seek to understand the extent to which changes in the IC parameters can produce changes in the influence function, and in particular, check if the function is Lipschitz. For this, we use the closed-form interpretation of the IC function as an expectation of an indicator term over a randomly drawn subset of edges from the network (see [2]). More specifically, the IC cascade process can be seen as activating a subset of edges in the network; since each edge can be activated at most once, the active edges can be seen as having been chosen apriori using independent Bernoulli draws. Consider a random subgraph of active edges obtained by choosing each edge $(u, v) \in E$ independently with probability $w_{uv}$. For a given subset of such edges $A \subseteq E$ and seed set $X \subseteq V$, let $\sigma_u(A, X)$ be an indicator function that evaluates to 1 if $u$ is reachable from a node in $X$ via edges in $A$ and 0 otherwise. Then the IC influence function can be written as an expectation of $\sigma$ over random draw of the subgraph:

$$F_u^{\mathbf{w}}(X) = \sum_{A \subseteq E} \prod_{(a,b) \in A} w_{ab} \prod_{(a,b) \notin A} (1 - w_{ab})\, \sigma_u(A, X). \tag{2}$$

While the above definition involves an exponential number of terms, it can be verified that the corresponding gradient is bounded, thus implying that the IC function is Lipschitz.[3]

**Lemma 3.** *Fix $X \subseteq V$. For any $\mathbf{w}, \mathbf{w}' \in \mathbb{R}^r$ with $\|\mathbf{w} - \mathbf{w}'\|_1 \le \epsilon$, $\left| F_u^{\mathbf{w}}(X) - F_u^{\mathbf{w}'}(X) \right| \le \epsilon$.*

This result tells us how small the parameter errors need to be to obtain accurate influence predictions and will be crucially used in our learnability results. Note that for the chain example in Section 2, this tells us that the errors need to be less than $1/n$ for meaningful influence predictions.

We are now ready to provide the PAC learning algorithm for the partial observation setting with sample $S = \{(X^1, Y^1), \ldots, (X^m, Y^m)\}$; we shall sketch the proof here. The full observation case is outlined in Section 4.1, where we shall make use of the a different approach based on local estimation. Let $F^{\mathbf{w}}$ denote the IC influence function with parameters $\mathbf{w}$. The algorithm that we consider for partial observation resorts to a maximum likelihood (ML) estimation of the (global) IC function. Let $\chi_u(Y) = \mathbf{1}(u \in Y)$. Define the (global) log-likelihood for a cascade $(X, Y)$ as:

$$\mathcal{L}(X, Y; \mathbf{w}) = \sum_{u=1}^{n} \chi_u(Y) \ln \left( F_u^{\mathbf{w}}(X) \right) + (1 - \chi_u(Y)) \ln \left( 1 - F_u^{\mathbf{w}}(X) \right),$$

The prescribed algorithm then solves the following optimization problem, and outputs an IC influence function $F^{\overline{\mathbf{w}}}$ from the solution $\overline{\mathbf{w}}$ obtained.

$$\max_{\mathbf{w} \in [\lambda, 1-\lambda]^r} \sum_{i=1}^{m} \mathcal{L}(X^i, Y^i; \mathbf{w}). \tag{3}$$

To provide learnability guarantees for the above ML based procedure, we construct a finite $\epsilon$-cover over the space of IC influence functions, i.e. show that the class can be approximated to a factor of $\epsilon$ (in the infinity norm sense) by a finite set of IC influence functions. We first construct an $\epsilon$-cover of size $O((r/\epsilon)^r)$ over the space of parameters $[\lambda, 1-\lambda]^r$, and use Lipschitzness to translate this to an $\epsilon$-cover of same size over the IC class. Following this, standard uniform convergence arguments [20] can be used to derive a sample complexity guarantee on the expected likelihood with a logarithmic dependence on the cover size; this then implies the desired learnability result w.r.t. $\ell_{sq}$:

**Lemma 4 (Sample complexity guarantee on the log-likelihood objective).** *Fix $\epsilon, \delta \in (0,1)$ and $m = \widetilde{O}(\epsilon^{-2} n^3 r)$. Let $\overline{\mathbf{w}}$ be the parameters obtained from ML estimation. Then w.p. $\geq 1 - \delta$,*

$$\sup_{\mathbf{w} \in [\lambda, 1-\lambda]^r} \mathbf{E}\left[\frac{1}{n}\mathcal{L}(X, Y; \mathbf{w})\right] - \mathbf{E}\left[\frac{1}{n}\mathcal{L}(X, Y; \overline{\mathbf{w}})\right] \leq \epsilon.$$

Compared to results for the LT model, the sample complexity in Theorem 2 has a square dependence on $1/\epsilon$. This is not surprising, as unlike the LT model, where the optimal 0-1 error is zero, the optimal squared error here is non-zero in general; in fact, there are standard sample complexity lower bound results that show that for similar settings, one cannot obtain a tighter bound in terms of $1/\epsilon$ [20].

We wish to also note that the approach of Du et al. (2014) for learning influence under partial observation [13] uses the same interpretation of the IC influence function as in Eq. (2), but rather than learning the parameters of the model, they seek to learn the weights on the individual indicator functions. Since there are exponentially many indicator terms, they resort to constructing approximations to the influence function, for which a strong technical condition needs to be satisfied; this condition need not however hold in most settings. In contrast, our result applies to general settings.

### 4.1  Efficient computation

**Partial observation.** The optimization problem in Eq. (3) that we need to solve for the partial observation case is non-convex in general. Of course, in practice, this can be solved approximately using gradient-based techniques, using sample-based gradient computations to deal with the exponential number of terms in the definition of $F^{\mathbf{w}}$ in the objective (see Appendix B.5).

**Full observation.** On the other hand, when training sample $S = \{(X^1, Y^1_{1:n}), \ldots, (X^m, Y^m_{1:n})\}$ contains fully observed cascades, we are able to show polynomial time learnability. For the LT model, we were assured of a set of parameters that would yield zero 0-1 error on the training sample, and hence the same procedure prescribed for partial information could be implemented under the full observation in polynomial time by reduction to local computations. This is not the case with the IC model, where we resort to the common approach of learning influence by estimating the model parameters through a *local* maximum likelihood (ML) estimation technique. This method is similar to the maximum likelihood procedure used in [9] for solving a different problem of recovering the structure of an unknown network from cascades. For the purpose of showing learnability, we find it sufficient to apply this procedure to only the first time step of the cascade.

Our analysis first provides guarantees on the estimated parameters, and uses the Lipschitz property in Lemma 3 to translate them to guarantees on the influence function. Since we now wish to give guarantees in the parameter space, we will require that there exists unique set of parameters that explains the IC cascade process; for this, we will need stricter assumptions. We assume that all edges have a minimum influence strength, and that even when all neighbors of a node $u$ are influenced in a time step, there is a small probability of $u$ not being influenced in the next step; we consider a specific seed distribution, where each node has a non-zero probability of (not) being a seed node.

**Assumption 1.** *Let $\mathbf{w}^*$ denote the parameters of the underlying IC model. Then there exists $\lambda \geq \gamma \in (0, 0.5)$ such that $w^*_{uv} \geq \lambda$ for all $(u,v) \in E$ and $\prod_{v \in N(u)}(1 - w_{uv}) \geq \gamma$ for all $u \in V$. Also, each node in $V$ is chosen independently in the initial seed set with probability $\kappa \in (0,1)$.*

We first define the local log-likelihood for given seed set $X$ and nodes $Y_1$ influenced at $t = 1$:

$$\mathcal{L}(X, Y_1; \boldsymbol{\beta}) = \sum_{u \notin X} \left[ \chi_u(Y_1) \ln\left(1 - \exp\left(-\sum_{v \in N(u) \cap X} \beta_{uv}\right)\right) - (1 - \chi_u(Y_1)) \sum_{v \in N(u) \cap X} \beta_{uv} \right],$$

where we have used log-transformed parameters $\beta_{uv} = -\ln(1 - w_{uv})$, so that the objective is concave in $\boldsymbol{\beta}$. The prescribed algorithm then solves the following maximization problem over all

parameters that satisfy Assumption 1 and constructs an IC influence function from the parameters.

$$\max_{\boldsymbol{\beta} \in \mathbb{R}_+^r} \sum_{i=1}^{m} \mathcal{L}(X^i, Y_1^i; \boldsymbol{\beta}) \quad \text{s.t.} \quad \forall (u,v) \in E, \ \beta_{uv} \geq \ln\left(\frac{1}{1-\lambda}\right), \ \forall u \in V, \sum_{v \in N(u)} \beta_{uv} \geq \ln\left(\frac{1}{\gamma}\right).$$

This problem breaks down into smaller convex problems and can be solved efficiently (see [9]).

**Proposition 5** (**PAC learnability under IC model with full observation**). *Under full observation and Assumption 1, the class of IC influence functions is PAC learnable in polynomial time through local ML estimation. The corresponding sample complexity is $\widetilde{O}\big(nr^3(\kappa^2(1-\kappa)^4\lambda^2\gamma^2\epsilon^2)^{-1}\big)$.*

The proof is provided in Appendix B.6 and proceeds through the following steps: (1) we use covering number arguments to show that the local log-likelihood for the estimated parameters is close to the optimal value; (2) we then show that under Assumption 1, the expected log-likelihood is *strongly concave*, which gives us that closeness to the true model parameters in terms of the likelihood also implies closeness to the true parameters in the parameter space; (3) we finally use the Lipschitz property in Lemma 3 to translate this to guarantees on the global influence function.

Note that the sample complexity here has a worse dependence on the number of edges $r$ compared to the partial observation case; this is due to the two-step approach of requiring guarantees on the individual parameters, and then transferring them to the influence function. The better dependence on the number of nodes $n$ is a consequence of estimating parameters locally. It would be interesting to see if tighter results can be obtained by using influence information from all time steps, and making different assumptions on the model parameters (e.g. correlation decay assumption in [9]).

## 5 The Voter model

Before closing, we sketch of our learnability results for the Voter model, where unlike previous models the graph is undirected (with self-loops). Here we shall be interested in learning influence for a fixed number of $K$ time steps as the cascades can be longer than $n$. With the squared loss again as the loss function, this problem almost immediately reduces to linear least squares regression.

Let $\mathbf{W} \in [0,1]^{n \times n}$ be a matrix of normalized edge weights with $W_{uv} = w_{uv}/\sum_{v \in N(u) \cup \{u\}} w_{uv}$ if $(u,v) \in E$ and 0 otherwise. Note that $\mathbf{W}$ can be seen as a one-step probability transition matrix. Then for an initial seed set $Z \subseteq V$, the probability of a node $u$ being influenced under this model after one time step can be verified to be $\mathbf{1}_u^\top \mathbf{W} \mathbf{1}_X$, where $\mathbf{1}_X \in \{0,1\}^n$ is a column vector containing 1 in entries corresponding to nodes in $X$, and 0 everywhere else. Similarly, for calculating the probability of a node $u$ being influenced after $K$ time steps, one can use the $K$-step transition matrix: $F_u(X) = \mathbf{1}_u^\top (\mathbf{W}^K) \mathbf{1}_X$. Now setting $\mathbf{b} = (\mathbf{W}^K)^\top \mathbf{1}_u$, we have $F_u(X) = \mathbf{b}^\top \mathbf{1}_X$ which is essentially a linear function parametrized by $n$ weights.

Thus learning influence in the Voter model (for fixed cascade length) can be posed as $n$ independent linear regression (one per node) with $n$ coefficients each. This can be solved in polynomial time even with partially observed data. We then have the following from standard results [20].

**Theorem 6** (**PAC learnability under Voter model**). *The class of influence functions under the Voter model is PAC learnable w.r.t. $\ell_{\text{sq}}$ in polynomial time and the sample complexity is $\widetilde{O}\big(\epsilon^{-2}n\big)$.*

## 6 Conclusion

We have established PAC learnability of some of the most celebrated models of influence in social networks. Our results point towards interesting connections between learning theory and the literature on influence in networks. Beyond the practical implications of the ability to learn influence functions from cascades, the fact that the main models of influence are PAC learnable, serves as further evidence of their potent modeling capabilities. It would be interesting to see if our results extend to generalizations of the LT and IC models, and to investigate sample complexity lower bounds.

**Acknowledgements.** Part of this work was carried out while HN was visiting Harvard as a part of a student visit under the Indo-US Joint Center for Advanced Research in Machine Learning, Game Theory & Optimization supported by the Indo-US Science & Technology Forum. HN thanks Kevin Murphy, Shivani Agarwal and Harish G. Ramaswamy for helpful discussions. YS and DP were supported by NSF grant CCF-1301976 and YS by CAREER CCF-1452961 and a Google Faculty Research Award.

## Footnotes

[1]In settings where the node thresholds are unknown, it is common to assume that they are chosen randomly by each node [2]. In our setup, the thresholds are parameters that need to be learned from cascades.

[2]By a strong signal, we mean a large positive/negative connection weight which will outweigh signals from other connections. Indeed such connections can be created when the weights are all bounded.

[3]In practice, IC influence functions can be computed through suitable sampling approaches. Also, note that a function class can be PAC learnable even if the individual functions cannot be computed efficiently.

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
