[Supplementary Material · learning-influence-supplementary.pdf]

# Learnability of Influence in Networks

# Appendix

## A  Proofs/Additional Material for Section 3

Recall that in the LT model, given a set of nodes $Z$ influenced at a previous time step, the local influence for a node $u$ (that has not been influenced so far) is given by $\mathbf{1}\big(\sum_{v \in N(u) \cap Z} w_{uv} \geq k_u\big)$, where $\mathbf{w}$ denotes a vector of edge weights and threshold parameters of the model. Let us use the notation $f_u^{\mathbf{w}}$ to denote this local influence function at $u$ (for ease of notation, we superscript $f$ with $\mathbf{w}$ though the function is defined on only a subset of indices of $\mathbf{w}$ relevant to $u$; note that the notation here is slightly different from the one in the main text). In the following we shall sometimes overload notation and allow influence functions to take boolean membership vector in $\{0,1\}^n$ as inputs (rather than sets) with each entry $u$ in the vector indicating whether node $u$ is present in the seed set. We shall use VCdim($\mathcal{F}$) to denote the VC-dimension of a (binary) function class $\mathcal{F}$. As before for any $Z \subseteq V$, we use the membership indicator $\chi_u(Z) = 1(u \in Z)$.

### A.1  Proof of Theorem 1

We provide here the proof for the partial observation setting. The proof for the full observation setting is the same for most part, except that the training set contains more fine-grained information about the set of nodes influenced in each time step of a cascade $Y_1, \ldots, Y_n$, which for the purpose of the proof can all be aggregated into one set: $\bigcup_{t=1}^{n} Y_t$. Recall that the learning algorithm here simply picks an influence function with zero training error. To show that this procedure PAC learns from the LT class, we start by bounding the VC-dimension of the class of LT influence functions $F_u^{\mathbf{w}}$ for a given node $u$. The proof then follows from standard uniform convergence arguments for function classes with finite VC-dimension.

**Lemma 7** (**VC-dimension of global LT influence functions**). *Fix node $u$. The class of all LT influence functions $F_u : 2^V \to \{0,1\}$ has a VC-dimension of at most $\widetilde{O}(r + n)$.*

*Proof.* We shall describe how the influence function $F_u^{\mathbf{w}}$ can be seen as a neural network and then extend classic results on the VC-dimension of neural networks to derive the VC-dimension of the class of all influence functions for node $u$. To build intuition, let us start with the neural network construction for a simpler setting where a node, once influenced, can influence its neighbors in all subsequent time steps. While the resulting influence process can now last for more than $n$ steps, we describe the construction for only $n$ time steps. We shall then extend this network to the setting considered in this paper where a node can influence its neighbors only once.

1. **Local influence as a two-layer neural network.** Recall that the (local) influence at a node $u$ is given by $f_u^{\mathbf{w}}(Z) = \mathbf{1}\big(\sum_{v \in N(u) \cap Z} w_{uv} \geq k_u\big)$. This function can be modeled as a linear (binary) classifier, or equivalently as a two-layer NN with linear threshold activations. Here the input layer contains a unit for each node in the social network and takes a binary value indicating whether the node is present in $Z$; the output layer contains a binary unit indicating whether $u$ is influenced after one time step; the connections between the two layers correspond to the edges between $u$ and other nodes; and the threshold term on the output unit is the threshold parameter $k_u$. Thus the first step of the influence process can be modeled using a NN with two $n$-node layers (the input layer takes information about the seed set, and output is a binary vector indicating which nodes got influenced).

2. **From local to global: the multilayer network.** The two-layer network can be extended to multiple time steps by replicating the second layer described above once for each step, along with the associated connections and thresholds. Additionally, let us add an edge from each node $u$ to itself with a weight that exceeds threshold $k_u$. Thus once a node $u$ is activated in a layer, it remains active thereafter. The LT influence function $F_u^{\mathbf{w}}$ (which outputs for any seed set, whether or not node $u$ will be influenced in the corresponding cascade) is given by the status of node $u$ in the last layer.

Thus $F_u^{\mathbf{w}}$ can be represented as a neural network with $n+1$ layers, with each layer containing $r+n$ parameters. If we ignore for a moment that the same parameters repeat across layers, an application of classic VC-dimension results for neural networks with $n(r+n)$ parameters, will give us that the VC-dimension of the class of all functions $F_u^{\mathbf{w}}$ for node $u$ is at most $O\big((n(r+n))\log(n(r+n))\big)$. However, using a more careful analysis one can get a tighter bound of $O((r+n)\log(r+n))$. This is because with each new layer with the same connection weights, the ability of a neural network to shatter a subset of points can only reduce.

To see this, let us denote by $F_{t,u}^{\mathbf{w}} : \{0,1\}^n \to \{0,1\}$ the function computed at node $u$ in layer $t+1$ for a given seed set encoded as binary vector in $\{0,1\}^n$ (recall that layer 1 is the input layer, and hence we only consider layer two onwards). Clearly, the function computed in the second layer $F_{1,u}^{\mathbf{w}}$ is equivalent to the local LT influence function $f_u^{\mathbf{w}}$, and that computed in the $(n+1)^{\text{th}}$ layer $F_{n,u}^{\mathbf{w}}$ is the required global influence function $F_u^{\mathbf{w}}$. Let $\mathcal{F}_{t,u}$ denote the class of all functions $F_{t,u}^{\mathbf{w}}$ under the LT model for different parameters $\mathbf{w} \in \mathbb{R}_+^{r+n}$. It is easy to see $\mathcal{F}_{1,u}$ is a class of linear binary classifiers with $r+1$ parameters, and hence we have from standard results that the $\text{VCdim}(\mathcal{F}_{1,u}) = r+1$. Similarly, $\mathcal{F}_{2,u}$ can be seen as a class of neural networks (linear threshold activations) with $O(r+n)$ parameters, and we have $\text{VCdim}(\mathcal{F}_{2,u}) = O((r+n)\ln(r+n))$. We shall now prove that $\text{VCdim}(\mathcal{F}_{t,u}) \leq O((r+n)\ln(r+n))$ for all $t \geq 3$. Consider a set of points $\{\mathbf{x}_1,\ldots,\mathbf{x}_N\} \subseteq \{0,1\}^n$ shattered by $\mathcal{F}_{t,k}, t \geq 3$. In other words, consider points such that

$$
\begin{aligned}
2^N &= |\{F_{t,u}^{\mathbf{w}}(\mathbf{x}_1),\ldots,F_{t,u}^{\mathbf{w}}(\mathbf{x}_N) \mid \mathbf{w} \in \mathbb{R}_+^{r+n}\}| \\
&= |\{F_{t-1,u}^{\mathbf{w}}(F_{1,1}^{\mathbf{w}}(\mathbf{x}_1),\ldots,F_{1,n}^{\mathbf{w}}(\mathbf{x}_1)),\ldots,F_{t-1,u}^{\mathbf{w}}(F_{1,1}^{\mathbf{w}}(\mathbf{x}_N),\ldots,F_{1,N}^{\mathbf{w}}(\mathbf{x}_N)) \mid \mathbf{w} \in \mathbb{R}_+^{r+n}\}| \\
&= |\{F_{t-1,u}^{\mathbf{w}}(\mathbf{z}_1),\ldots,F_{t-1,u}^{\mathbf{w}}(\mathbf{z}_N) \mid \mathbf{w} \in \mathbb{R}_+^{r+n}\}|,
\end{aligned}
$$

where $\mathbf{z}_1 = [F_{1,1}^{\mathbf{w}}(\mathbf{x}_1),\ldots,F_{1,n}^{\mathbf{w}}(\mathbf{x}_1)]$, ..., $\mathbf{z}_N = [F_{1,1}^{\mathbf{w}}(\mathbf{x}_N),\ldots,F_{1,n}^{\mathbf{w}}(\mathbf{x}_N)] \in \{0,1\}^n$. Since $|\{F_{t-1,u}^{\mathbf{w}}(\mathbf{z}_1),\ldots,F_{t-1,u}^{\mathbf{w}}(\mathbf{z}_N) \mid \mathbf{w} \in \mathbb{R}_+^{r+n}\}| = 2^N$, it is necessarily the case that $\mathbf{z}_1,\ldots,\mathbf{z}_N$ are different (if not, not all binary assignments in $\{0,1\}^N$ can be realized). This implies that the set of points $\{\mathbf{z}_1,\ldots,\mathbf{z}_N\}$ is shattered by $\mathcal{F}_{t-1,u}$. Thus for any set of points of a given size shattered by $\mathcal{F}_{t,u}$, there exists a set of points of the same size shattered by $\mathcal{F}_{t-1,u}$. This gives us that the VC-dimension of $\mathcal{F}_{t,u}$ is no greater than the VC-dimension of $\mathcal{F}_{t-1,u}$, i.e. $\text{VCdim}(\mathcal{F}_{t,u}) \leq \text{VCdim}(\mathcal{F}_{t-1,u})$ for all $t \geq 3$; applying this argument recursively, we have $\text{VCdim}(\mathcal{F}_{t,u}) \leq \text{VCdim}(\mathcal{F}_{2,u}) = O((r+n)\ln(r+n))$. Thus $\text{VCdim}(\mathcal{F}_{n,u}) \leq O((r+n)\ln(r+n))$.

The above result is for a simpler setting where a node, once influenced, can influence its neighbors in all subsequent time steps. In the setting that we consider in this paper, a node gets influenced only by neighbors who were influenced in the previous time step, and moreover, a node cannot get influenced more than once during a cascade. To incorporate this additional constraint in the neural network structure, we introduce an additional binary unit $u'$ for each node $u$ in a layer, which will record whether node $u$ was influenced in previous time steps. In particular, whenever node $u$ is influenced in a layer, a strong positive signal is sent to activate $u'$ in the next layer, which in turn will send out strong negative signals to ensure $u$ is never activated in subsequent layer; we use additional connections to ensure that $u'$ remains active there after. In the resulting neural network, a node $u$ is activated in layer $t+1$ whenever $u$ is influenced exactly at time step $t$; a node is never activated again in subsequent time steps (see Figure 1). Hence, if $F_{t,u}^{\mathbf{w}} : 2^V \to \{0,1\}$ is the function computed at node $u$ in layer $t+1$, then the global LT influence function is given by $F_u^{\mathbf{w}}(X) = \sum_{t=1}^n F_{t,u}^{\mathbf{w}}(X)$. It can be verified that $F_u^{\mathbf{w}}$ can also be modeled as a neural network with $n+1$ layers and $r+n$ parameters. The same analysis used above can be retraced to show that the VC-dimension of all functions $F_u^{\mathbf{w}}$ for node $u$ is $O((r+n)\ln(r+n))$.[4]   $\square$

We are now ready to prove our theorem.

*Proof of Theorem 1.* As before, $\mu$ denotes the distribution over the initial seed sets and $\mathbf{w}^*$ denotes the parameters of the underlying model; note that in this setting, $\inf_{\mathbf{w}\in\mathbb{R}_+^{r+n}} \text{err}^{0\text{-}1}[F^{\mathbf{w}}] = \text{err}^{0\text{-}1}[F^{\mathbf{w}^*}] = 0$; this also means that $\mathbf{E}\big[\mathbf{1}(\chi_u(Y)) \neq F_u^{\mathbf{w}^*}(X)\big] = 0 \ \forall u \in V$. Also, let $\overline{\mathbf{w}}$ denote the parameters obtained from Eq. (1); since $\overline{\mathbf{w}}$ minimizes the training error, we have for all $u$,

$\frac{1}{m}\sum_{i=1}^{m}\mathbf{1}\big(\chi_u(Y^i)\neq F_u^{\overline{\mathbf{w}}}(X^i)\big)=0$. We also know from Lemma 7, that for each $u$, the class of all influence functions $F_u^{\mathbf{w}}:2^V\to\{0,1\}$ has a VC-dimension of $O((r+n)\ln(r+n))$. We can then use standard VC-dimension based learnability results for empirical risk minimization in settings where there is a function in the given function class that correctly labels all examples [20]. In particular, we have for any $\epsilon,\delta\in(0,1)$, and $m=O\bigg(\dfrac{(r+n)\ln(r+n)\ln(1/\epsilon)+\ln(1/\delta)}{\epsilon}\bigg)$, with probability at least $1-\delta$ (over draw of the training sample), $\mathbf{E}\big[\mathbf{1}(\chi_u(Y))\neq F_u^{\overline{\mathbf{w}}}(X)\big]\leq\epsilon$. Taking a union bound over all of $n$ nodes now gives us that when $m=O\bigg(\dfrac{(r+n)\ln(r+n)\ln(1/\epsilon)+\ln(n/\delta)}{\epsilon}\bigg)$, with probability at least $1-\delta$,

$$\mathrm{err}^{\text{0-1}}[F^{\overline{\mathbf{w}}}] \;=\; \frac{1}{n}\sum_{u=1}^{n}\mathbf{E}\big[\mathbf{1}\big(\chi_u(Y)\neq F_u^{\overline{\mathbf{w}}}(X)\big)\big] \;\leq\; \epsilon.$$

This completes the proof. $\hfill\square$

### A.2 Formulating the learning problem as a LP under full observation

Let $n_u=|N(u)|$. Under full observation, the problem of obtaining parameters for which the local prediction error is zero for a given node $u\in V$ can be equivalently framed as the following linear program. Here the optimization is over $\mathbf{w}_u\in\mathbb{R}_+^{n_u+1}$ and over slack variables $\xi_{i,t}$ for each cascade $i$ and time step $t$, subject to 'margin' constraints enforcing that the predicted influence status agrees with the true status of a node for each time step and training cascade.

$$\min_{\mathbf{w}_u\in\mathbb{R}_+^{n_u+1},\,\xi_{i,t}\geq 0}\;\sum_{i=1}^{m}\sum_{t=1}^{n}\xi_{i,t}$$

$$\big(2\chi_u(Y_t^i)-1\big)\Bigg(\sum_{v\in N(u)\cap Y_{t-1}^i}w_{uv}-k_u\Bigg)\;\geq\;1-\xi_{i,t},\;\;\forall\,i\in[m],\,t\in[T].$$

Let $\mathbf{w}^*\in\mathbb{R}_+^{r+n}$ denote the parameters of the LT model from which the training sample was generated and $\mathbf{w}_u^*\in\mathbb{R}_+^{n_u+1}$ denote the parameters corresponding to node $u\in V$. Then $\mathbf{w}_u^*$ yields zero prediction error for node $u$; this also means that there exists a scaled version of $\mathbf{w}_u^*$ which yields optimal slack values $\xi_{i,t}^*=0$ in the above problem. Clearly, solving the above LP will recover a scaled version of $\mathbf{w}_u^*$.

## B  Proofs/Additional Details for Section 4

Here, given a set of nodes $Z\subseteq V$ influenced at a time step, the probability of node $u$ (that has not been influenced so far) being influenced in the next time step is $f_u^{\mathbf{w}}(Z)=1-\prod_{v\in N(u)\cap Z}(1-w_{uv})$. As before, for any $Z\subseteq V$, $\chi_u(Z)=\mathbf{1}(u\in Z)$.

### B.1  Proof of Theorem 2

We deal with the partial observation setting here. The full observation case is handled in the proof of Proposition 5 in Section B.6. The algorithm prescribed for the partial observation setting is a global maximum likelihood estimation described in Section 4; the specific optimization problem that needs to be solved is given in Eq. (3).

We start with an outline of the proof:

- We first show that the IC influence function $F_u^{\mathbf{w}}$ is 1-Lipschitz w.r.t. the $L_1$ norm (i.e. bounded changes in parameters only produce bounded changes in the function values). This was stated in Lemma 3 in the main text (restated below).

  **Lemma 3.** (**Lipschitzness of IC influence function w.r.t. $L_1$ norm**). Fix $X\subseteq V$. For any $\mathbf{w},\mathbf{w}'\in\mathbb{R}^r$ with $\|\mathbf{w}-\mathbf{w}'\|_1\leq\epsilon$, $\big|F_u^{\mathbf{w}}(X)-F_u^{\mathbf{w}'}(X)\big|\leq\epsilon$.

*Proof.* See Section B.2. □

- We then establish an $\epsilon$-cover over the space of parameters $[0,1]^r$ and translate this using the above Lipschitz property to a $\epsilon$-cover over the space of IC influence functions, thus obtaining a bound on the covering number of this space.

  **Lemma 8** (**Covering number of IC influence functions**). *The $L_\infty$ covering number of the class of all IC influence functions $F_u$ for radius $\epsilon$ is $O((r/\epsilon)^r)$.*

  *Proof.* See Section B.3. □

- Next, we appeal to standard uniform convergence arguments based on covering numbers [20] to bound the difference between the expected log-likelihood for the estimated parameters $\overline{\mathbf{w}}$ and that for the true parameters $\mathbf{w}^*$. This was stated in Lemma 4 in the main text (restated below).

  **Lemma 4** (**Sample complexity guarantee on the log-likelihood objective**). *Fix $\epsilon, \delta \in (0,1)$ and $m = \widetilde{O}(\epsilon^{-2}n^3r)$. Let $\overline{\mathbf{w}}$ be the parameters obtained from global ML estimation. With probability at least $1 - \delta$ (over draw of the training sample), we have that*

$$\sup_{\mathbf{w} \in [\lambda, 1-\lambda]^r} \mathbf{E}\left[\frac{1}{n}\mathcal{L}(X, Y; \mathbf{w})\right] - \mathbf{E}\left[\frac{1}{n}\mathcal{L}(X, Y; \overline{\mathbf{w}})\right] \le \epsilon.$$

  *Proof.* See Section B.4. □

- Finally, the above guarantee is translated into a bound on the difference between the expected squared error for $\overline{\mathbf{w}}$ and that for $\mathbf{w}^*$, as we shall see below.

*Proof of Theorem 2.* For PAC learnability in this setting, we need to show that $\mathrm{err}^{\mathrm{sq}}\big[F^{\overline{\mathbf{w}}}\big] - \inf_{\mathbf{w} \in \mathbb{R}_+^r} \mathrm{err}^{\mathrm{sq}}\big[F^{\mathbf{w}}\big] = \mathrm{err}^{\mathrm{sq}}\big[F^{\overline{\mathbf{w}}}\big] - \mathrm{err}^{\mathrm{sq}}\big[F^{\mathbf{w}^*}\big]$ can be made arbitrarily small w.h.p. Expanding this, we have

$$
\begin{aligned}
\mathrm{err}^{\mathrm{sq}}&\big[F^{\overline{\mathbf{w}}}\big] - \mathrm{err}^{\mathrm{sq}}\big[F^{\mathbf{w}^*}\big] \\
&= \mathbf{E}_{X,Y}\big[\ell_{\mathrm{sq}}\big(Y, F^{\overline{\mathbf{w}}}(X)\big)\big] - \mathbf{E}_{X,Y}\big[\ell_{\mathrm{sq}}\big(Y, F^{\mathbf{w}^*}(X)\big)\big] \\
&= \mathbf{E}_{X,Y}\big[\ell_{\mathrm{sq}}\big(Y, F^{\overline{\mathbf{w}}}(X)\big) - \ell_{\mathrm{sq}}\big(Y, F^{\mathbf{w}^*}(X)\big)\big], \\
&= \frac{1}{n}\sum_{u=1}^{n} \mathbf{E}_{X,Y}\Big[\chi_u(Y)(1 - F_u^{\overline{\mathbf{w}}}(X))^2 + (1 - \chi_u(Y))F_u^{\overline{\mathbf{w}}}(X)^2 \\
&\qquad\qquad\qquad - \chi_u(Y)(1 - F_u^{\mathbf{w}^*}(X))^2 - (1 - \chi_u(Y))F_u^{\mathbf{w}^*}(X)^2\Big] \\
&= \frac{1}{n}\sum_{u=1}^{n} \mathbf{E}_X\Big[\mathbf{E}_{Y|X}\Big[\chi_u(Y)(1 - F_u^{\overline{\mathbf{w}}}(X))^2 + (1 - \chi_u(Y))F_u^{\overline{\mathbf{w}}}(X)^2 \\
&\qquad\qquad\qquad - \chi_u(Y)(1 - F_u^{\mathbf{w}^*}(X))^2 - (1 - \chi_u(Y))F_u^{\mathbf{w}^*}(X)^2\Big]\Big] \\
&= \frac{1}{n}\sum_{u=1}^{n} \mathbf{E}_X\Big[F_u^{\mathbf{w}^*}(X)(1 - F_u^{\overline{\mathbf{w}}}(X))^2 + (1 - F_u^{\mathbf{w}^*}(X))F_u^{\overline{\mathbf{w}}}(X)^2 \\
&\qquad\qquad\qquad - F_u^{\mathbf{w}^*}(X)(1 - F_u^{\mathbf{w}^*}(X))^2 - (1 - F_u^{\mathbf{w}^*}(X))F_u^{\mathbf{w}^*}(X)^2\Big] \\
&= \frac{1}{n}\sum_{u=1}^{n} \mathbf{E}_X\Big[\big(F_u^{\overline{\mathbf{w}}}(X) - F_u^{\overline{\mathbf{w}}}(X)\big)^2\Big], \qquad\qquad (4)
\end{aligned}
$$

where the fifth step follows from the fact that for any $X$, $\mathbf{E}_Y[\chi_u(Y)\,|\,X] = F_u^{\mathbf{w}^*}(X)$.

We already have from Lemma 4 that when the number of training examples $m = \widetilde{O}(\epsilon^{-2}n^3r)$, we have with probability at least $1 - \delta$ (over draw of training sample),

$$\mathbf{E}_{X,Y}\left[\frac{1}{n}\mathcal{L}(X, Y; \mathbf{w}^*)\right] - \mathbf{E}_{X,Y}\left[\frac{1}{n}\mathcal{L}(X, Y; \overline{\mathbf{w}})\right] \le \epsilon.$$

Expanding the left-hand side of the above inequality,

$$\frac{1}{n}\mathbf{E}_{X,Y}\big[\mathcal{L}(X,Y;\mathbf{w}^*) - \mathcal{L}(X,Y;\overline{\mathbf{w}})\big]$$

$$= \frac{1}{n}\sum_{u=1}^{n}\mathbf{E}_{X,Y}\big[\chi_u(Y)\ln(F_u^{\mathbf{w}^*}(X)) + (1-\chi_u(Y))\ln(1-F_u^{\mathbf{w}^*}(X))$$

$$- \chi_u(Y)\ln(F_u^{\overline{\mathbf{w}}}(X)) - (1-\chi_u(Y))\ln(1-F_u^{\overline{\mathbf{w}}}(X))\big]$$

$$= \frac{1}{n}\sum_{u=1}^{n}\mathbf{E}_{X}\big[F_u^{\mathbf{w}^*}(X)\ln(F_u^{\mathbf{w}^*}(X)) + (1-F_u^{\mathbf{w}^*}(X))\ln(1-F_u^{\mathbf{w}^*}(X))$$

$$- F_u^{\mathbf{w}^*}(X)\ln(F_u^{\overline{\mathbf{w}}}(X)) - (1-F_u^{\mathbf{w}^*}(X))\ln(1-F_u^{\overline{\mathbf{w}}}(X))\big]$$

$$= \frac{1}{n}\sum_{u=1}^{n}\mathbf{E}_{X}\big[L_{\log}\big(F_u^{\mathbf{w}^*}(X), F_u^{\mathbf{w}^*}(X)\big) - L_{\log}\big(F_u^{\mathbf{w}^*}(X), F_u^{\overline{\mathbf{w}}}(X)\big)\big]$$

$$\geq \frac{1}{n}\sum_{u=1}^{n}\mathbf{E}_{X}\big[2\big(F_u^{\mathbf{w}^*}(X) - F_u^{\overline{\mathbf{w}}}(X)\big)^2\big],$$

where the second equality follows from $E[\chi_u(Y)|X] = F_u^{\mathbf{w}^*}(X)$; in the second-last step last, we denote for any $\eta, \eta' \in [0,1]$, $L_{\log}(\eta',\eta) = \eta'\ln(\eta) + (1-\eta')\ln(1-\eta)$; the last step follows from the fact $L_{\log}(\eta',\eta') - L_{\log}(\eta',\eta) \geq 2(\eta-\eta')^2$ (this is easy to show; see e.g. Eq. (12) in [22]). Plugging this back into Eq. (4), the above implies that with probability at least $1-\delta$,

$$\operatorname{err}^{\text{sq}}\big[F^{\overline{\mathbf{w}}}\big] - \inf_{\mathbf{w}\in[\lambda,1-\lambda]}\operatorname{err}^{\text{sq}}\big[F^{\mathbf{w}}\big] \leq 0.5\epsilon, \qquad \text{as desired.}$$

$\square$

## B.2 Proof of Lemma 3

*Proof.* We bound the $L_\infty$ norm of the gradient of $F_u^{\mathbf{w}}$ by 1, which would imply that the function is 1-Lipschitz w.r.t. the $L_1$ norm. We have from Eq. (2), for any $(c,d) \in E$

$$\left|\frac{\partial F_u^{\mathbf{w}}(Z)}{\partial w_{cd}}\right|$$

$$= \left|\frac{\partial}{\partial w_{cd}}\left[w_{cd}\sum_{A\subseteq E\backslash\{(c,d)\}}\prod_{(a,b)\in A}w_{ab}\prod_{(a,b)\notin A,(a,b)\neq(c,d)}(1-w_{ab})\sigma_u(A\cup\{(c,d)\},Z)\right.\right.$$

$$\left.\left. + (1-w_{cd})\sum_{A\subseteq E\backslash\{(c,d)\}}\prod_{(a,b)\in A}w_{ab}\prod_{(a,b)\notin A,(a,b)\neq(c,d)}(1-w_{ab})\sigma_u(A,Z)\right]\right|$$

$$= \left|\sum_{A\subseteq E\backslash\{(c,d)\}}\prod_{(a,b)\in A}w_{ab}\prod_{(a,b)\notin A,(a,b)\neq(c,d)}(1-w_{ab})\sigma_u(A\cup\{(c,d)\},Z)\right.$$

$$\left. - \sum_{A\subseteq E\backslash\{(c,d)\}}\prod_{(a,b)\in A}w_{ab}\prod_{(a,b)\notin A,(a,b)\neq(c,d)}(1-w_{ab})\sigma_u(A,Z)\right|$$

$$\leq \left|\sum_{A\subseteq E\backslash\{(c,d)\}}\prod_{(a,b)\in A}w_{ab}\prod_{(a,b)\notin A,(a,b)\neq(c,d)}(1-w_{ab})\right|$$

$$= 1,$$

where the second last step follows from $0 \leq \sigma_u(A,Z) \leq 1$. Clearly $\|\nabla_{\mathbf{w}}F_u^{\mathbf{w}}(X)\|_\infty \leq 1$, which completes the proof of Lipschitzness of $F_u^{\mathbf{w}}$. $\square$

## B.3 Proof of Lemma 8

*Proof.* Note that the space of all parameters $\mathbf{w} \in [0,1]^r$ is bounded and can be covered by $(r/\epsilon)^r$ $L_1$-balls of radius $\epsilon$. Further, from the above lemma we known that $F_u^{\mathbf{w}}$ is 1-Lipschitz w.r.t. the $L_1$ norm; we then have for any $\mathbf{w}, \mathbf{w}' \in [0,1]^r$:

$$\max_{Z\subseteq V}\left|F_u^{\mathbf{w}}(Z) - F_u^{\mathbf{w}'}(Z)\right| \leq \|\mathbf{w} - \mathbf{w}'\|_1.$$

This says that if the parameters of two influence functions are separated by a distance of $\epsilon$ in the $L_1$ space, the influence functions are also within an $L_\infty$ distance of $\epsilon$ from each other. Clearly, an $L_1$ cover of radius $\epsilon$ over the parameter space can be translated to a $L_\infty$ cover of the over the space of all influence functions for node $u$. In particular, if the parameter space is covered by $R$ $L_1$-balls of radius $\epsilon$ and centers $\mathbf{w}_1, \ldots, \mathbf{w}_R$, then the influence functions $F^{\mathbf{w}_1}, \ldots, F^{\mathbf{w}_R}$ form a $L_\infty$ cover of the space of influence functions, with the same radius. Thus the number of $L_\infty$-balls of radius $\epsilon$ required to cover the space of influence functions is at most $O\big((r/\epsilon)^r\big)$. $\qquad\square$

### B.4 Proof of Lemma 4

The proof makes use of standard covering number based uniform convergence result for empirical risk minimization (or equivalently for log-likelihood maximization) over a real-valued function class [20]. To apply these standard results, we must ensure the log-likelihood is bounded and Lipschitz. We shall first establish this.

We once again use $n_u = |N(u)|$. Define for any $Z \subseteq V$, $y \in \{0, 1\}$, $\mathbf{w} \in [\lambda, 1 - \lambda]^r$ and $u \in V$, a function: $g_u(Z, y; \mathbf{w}) = y \ln \big(F_u^{\mathbf{w}}(Z)\big) + (1 - y) \ln \big(1 - F_u^{\mathbf{w}}(Z)\big)$. Note that for a cascade $(X, Y)$, $\mathcal{L}(X, Y; \mathbf{w}) = \frac{1}{n} \sum_{u=1}^{n} g_u(X, \chi_u(Y); \mathbf{w})$. In the following lemma, whenever we refer to a subset $Z \subseteq V$ in the context of a node $u$, *we shall assume that $u \notin Z$ and that there exists a path in the graph from a node in $Z$ to $u$*; cases where this assumption fails can be easily handled, but have been ignored here to make the proof easier to follow. Below, we show that $g_u$ is bounded and Lipschitz for any $u$.

**Lemma 9** (**Boundedness and Lipschitz continuity of log-likelihood function**). *Fix parameters* $\mathbf{w} \in [\lambda, 1 - \lambda]^r$. *Then*

1. $\lambda^n \leq F_u^{\mathbf{w}}(Z) \leq 1 - \lambda^n$.

2. $|g_u(Z, y; \mathbf{w})| \leq n \ln(1/\lambda)$.

3. $g_u(Z, y; \mathbf{w})$ *is $1/\lambda^n$-Lipschitz in $\mathbf{w}$ w.r.t. the $L_1$ norm.*

*Proof.*
1. Starting with the lower bound, recall the interpretation of the IC influence function in Eq. (2) as an expectation of an indicator term over random draw of a subgraph. From this interpretation, it is clear that the probability of node $u$ being influenced is at least the probability that all edges in a path from a node in $Z$ to $u$ get activated. Since the minimum probability on any edge is $\lambda$ and the length of any path can be at most $n$, we have $F_u^{\mathbf{w}}(Z) \geq \lambda^n$. For the upper bound, note that the probability of $u$ *not* being influenced in any of $n$ time steps for a seed set $Z$ is at least the probability that none of the neighbors of $u$ ever influenced it (i.e. none of the incoming edges incident on $u$ got activated): $\prod_{v \in N(u)} (1 - w_{uv}) \geq (1 - (1 - \lambda))^{n_u} \geq \lambda^n$. Hence $F_u^{\mathbf{w}}(Z) \leq 1 - \lambda^n$.

2. $|g_u(Z, y; \mathbf{w})| = |y \ln \big(F_u^{\mathbf{w}}(Z)\big) + (1 - y) \ln \big(1 - F_u^{\mathbf{w}}(Z)\big)| \leq |y \ln(\lambda^n) + (1 - y) \ln(\lambda^n)| \leq n \ln(\lambda)$ (from lower and upper bounds on $F_u^{\mathbf{w}}$ derived above and from $\lambda < 1$).

3. To show Lipschitzness of $g_u$ w.r.t. $L_1$ norm, we bound the $L_\infty$ norm of its gradient. In particular, $g_u(Z, y; \mathbf{w}) = y \ln \big(1 - F_u^{\mathbf{w}}(Z)\big) - (1 - y) \ln \big(F_u^{\mathbf{w}}(Z)\big)$ and

$$\nabla_{\mathbf{w}} g_u(Z, y; \mathbf{w}) = \left[\frac{y}{F_u^{\mathbf{w}}(Z)} - \frac{1 - y}{1 - F_u^{\mathbf{w}}(Z)}\right] \nabla_{\mathbf{w}} F_u^{\mathbf{w}}(Z).$$

Since $1 - \lambda^n \geq F_u^{\mathbf{w}}(Z) \geq \lambda^n$, we have

$$\left|\frac{y}{F_u^{\mathbf{w}}(Z)} - \frac{1 - y}{1 - F_u^{\mathbf{w}}(Z)}\right| \leq \frac{1}{\lambda^n}.$$

In addition, from the Lipschitz property of the IC influence function in Lemma 3, we know its gradient norm is bounded by 1,

$$\big\|\nabla_{\mathbf{w}} g_u(Z, y; \mathbf{w})\big\|_\infty \leq \frac{1}{\lambda^n} \|\nabla_{\mathbf{w}} F_u^{\mathbf{w}}(Z)\|_\infty \leq \frac{1}{\lambda^n}(1).$$

Hence $g_u$ is $1/\lambda^n$-Lipschitz in $\mathbf{w}$ w.r.t. the $L_1$ norm.

$\square$

*Proof of Lemma 4.* Let $\overline{\mathbf{w}}$ be the parameters obtained by solving Eq. (3). Similarly, let $\mathbf{w}^* \in [\lambda, 1 - \lambda]^r$ be the the underlying model parameters. Since the cascades are generated from an IC model defined by $\mathbf{w}^*$, one can verify that maximizing the expected log-likelihood $\mathbf{E}_{X,Y}\big[\mathcal{L}(X, Y; \mathbf{w})\big]$ over all $\mathbf{w} \in [\lambda, 1 - \lambda]^r$ yields $\mathbf{w}^*$. As mentioned above, the proof involves an application of standard covering number based uniform convergence arguments[20]; we shall make use of the covering number result in Lemma 8 and the Lipschitzness and boundedness of the likelihood shown in Lemma 9.

First, let us write the likelihood objective in Eq. (3) in terms of $g_u$.

$$\frac{1}{mn} \sum_{i=1}^m \mathcal{L}(X^i, Y^i; \mathbf{w}) = \frac{1}{n} \sum_{u=1}^n \underbrace{\frac{1}{m} \sum_{i=1}^m g_u(X, \chi_u(Y^i); \mathbf{w})}_{\widehat{G}_u(\mathbf{w})}. \tag{5}$$

Similarly, the expected log-likelihood can be written as

$$\frac{1}{n} \mathbf{E}_{X,Y}\big[\mathcal{L}(X, Y; \mathbf{w})\big] = \frac{1}{n} \sum_{u=1}^n \underbrace{\mathbf{E}_{X,Y}\big[g_u(X, \chi_u(Y); \mathbf{w})\big]}_{G_u(\mathbf{w})}. \tag{6}$$

We proceed by bounding the difference between the expected and empirical log-likelihood objective for any model vector, and use this to bound the difference between the optimal likelihood and the likelihood value of $\overline{\mathbf{w}}$.

We know from Lemma 9 that $g_u$ is bounded by $n \ln(1/\lambda)$ and is $1/\lambda^n$-Lipschitz in $\mathbf{w}$. We can then invoke standard uniform convergence arguments based on the covering number result in Lemma 8, followed by a union bound over all nodes, to bound the difference between $G_u$ and $\widehat{G}_u$. In particular, when $m = O\left(n^2 \ln(1/\lambda)^2 \dfrac{r \ln(r/\epsilon) + nr \ln(1/\lambda) + \ln(n/\delta)}{\epsilon^2}\right)$, with probability at least $1 - \delta$ (over draw of training sample), for each $u \in V$ and all $\mathbf{w} \in [\lambda, 1 - \lambda]^r$,

$$|G_u(\mathbf{w}) - \widehat{G}_u(\mathbf{w})| \leq \epsilon/2.$$

Substituting this back into Eq. (5) and (6), gives us with probability at least $1 - \delta$, for all $\mathbf{w} \in [\lambda, 1 - \lambda]^r$,

$$\left|\mathbf{E}_{X,Y}\left[\frac{1}{n}\mathcal{L}(X, Y; \mathbf{w})\right] - \frac{1}{mn} \sum_{i=1}^m \mathcal{L}(X^i, Y^i; \mathbf{w})\right| \leq \frac{1}{n} \sum_{u=1}^n \epsilon/2 = \epsilon/2. \tag{7}$$

The above bound will then allow us to in turn bound the difference between the optimal log-likelihood and the log-likelihood of $\overline{\mathbf{w}}$, as shown below:

$$\sup_{\mathbf{w} \in [\lambda, 1 - \lambda]^r} \mathbf{E}_{X,Y}\left[\frac{1}{n}\mathcal{L}(X, Y; \mathbf{w})\right] - \mathbf{E}_{X,Y}\left[\frac{1}{n}\mathcal{L}(X, Y; \overline{\mathbf{w}})\right]$$

$$= \mathbf{E}_{X,Y}\left[\frac{1}{n}\mathcal{L}(X, Y; \mathbf{w}^*)\right] - \mathbf{E}_{X,Y}\left[\frac{1}{n}\mathcal{L}(X, Y; \overline{\mathbf{w}})\right]$$

$$= \mathbf{E}_{X,Y}\left[\frac{1}{n}\mathcal{L}(X, Y; \mathbf{w}^*)\right] - \frac{1}{mn} \sum_{i=1}^m \mathcal{L}(X^i, Y^i; \overline{\mathbf{w}})$$

$$\qquad + \frac{1}{mn} \sum_{i=1}^m \mathcal{L}(X^i, Y^i; \overline{\mathbf{w}}) - \mathbf{E}_{X,Y}\left[\frac{1}{n}\mathcal{L}(X, Y; \overline{\mathbf{w}})\right]$$

$$\leq \left[\mathbf{E}_{X,Y}\left[\frac{1}{n}\mathcal{L}(X, Y; \mathbf{w}^*)\right] - \frac{1}{mn} \sum_{i=1}^m \mathcal{L}(X^i, Y^i; \mathbf{w}^*)\right]$$

$$\qquad + \left[\frac{1}{mn} \sum_{i=1}^m \mathcal{L}(X^i, Y^i; \overline{\mathbf{w}}) - \mathbf{E}_{X,Y}\left[\frac{1}{n}\mathcal{L}(X, Y; \overline{\mathbf{w}})\right]\right]$$

$$\leq \quad \epsilon/2 + \epsilon/2 \quad = \quad \epsilon,$$

where the second-last step uses the fact that $\overline{\mathbf{w}}$ is the empirical maximizer of the log-likelihood, and the last step follows from Eq. (7). This completes the proof. $\qquad\square$

### B.5 Gradient Computation for Likelihood in Eq. (3)

We prescribe that the optimization problem in Eq. (3) be solved using a suitable gradient-based solver. We describe here how the gradient for the objective can be computed approximately by sampling subgraphs from $G$. In particular, for any $Z, Y \subseteq V$, and $(c, d) \in E$

$$
\begin{aligned}
\frac{\partial \mathcal{L}(Z, Y; \mathbf{w})}{\partial w_{cd}} &= \frac{\partial}{\partial w_{cd}} \left[ \sum_{u=1}^{n} \chi_u(Y) \ln \left( F_u^{\mathbf{w}}(X) \right) + (1 - \chi_u(Y)) \ln \left( 1 - F_u^{\mathbf{w}} \right) \right] \\
&= \sum_{u=1}^{n} \left[ \frac{\chi_u(Y)}{F_u^{\mathbf{w}}(X)} - \frac{1 - \chi_u(Y)}{1 - F_u^{\mathbf{w}}(X)} \right] \frac{\partial F_u^{\mathbf{w}}(Z)}{\partial w_{cd}}.
\end{aligned}
$$

Further,

$$
\begin{aligned}
\frac{\partial F_u^{\mathbf{w}}(Z)}{\partial w_{cd}} &= \frac{\partial}{\partial w_{cd}} \Bigg[ w_{cd} \sum_{A \subseteq E \setminus \{(c,d)\}} \prod_{(a,b) \in A} w_{ab} \prod_{(a,b) \notin A,\, (a,b) \neq (c,d)} (1 - w_{ab}) \sigma_u(A \cup \{(c,d)\}, Z) \\
&\qquad\qquad + (1 - w_{cd}) \sum_{A \subseteq E \setminus \{(c,d)\}} \prod_{(a,b) \in A} w_{ab} \prod_{(a,b) \notin A,\, (a,b) \neq (c,d)} (1 - w_{ab}) \sigma_u(A, Z) \Bigg] \\
&= \sum_{A \subseteq E \setminus \{(c,d)\}} \prod_{(a,b) \in A} w_{ab} \prod_{(a,b) \notin A,\, (a,b) \neq (c,d)} (1 - w_{ab}) \sigma_u(A \cup \{(c,d)\}, Z) \\
&\qquad - \sum_{A \subseteq E \setminus \{(c,d)\}} \prod_{(a,b) \in A} w_{ab} \prod_{(a,b) \notin A,\, (a,b) \neq (c,d)} (1 - w_{ab}) \sigma_u(A, Z) \qquad (8) \\
&= \underbrace{\sum_{A \subseteq E \setminus \{(c,d)\}} \mathbf{P}_{(c,d)}[A]\, \sigma_u(A \cup \{(c,d)\}, Z)}_{\text{term}_1} - \underbrace{\sum_{A \subseteq E \setminus \{(c,d)\}} \mathbf{P}_{(c,d)}[A]\, \sigma_u(A, Z)}_{\text{term}_2},
\end{aligned}
$$

where $\mathbf{P}_{(c,d)}[A]$ denotes the probability of sampling the edge subset $A$ when each edge $(u, v) \neq (c, d)$ is chosen independently with probability $w_{uv}$. Thus to compute the gradient of optimization objective in Eq. (3), we will need to evaluate the values of $F_u^{\mathbf{w}}$, $\text{term}_1$ and $\text{term}_2$ for every node $u$ and training example. While each of these involve a summation over an exponential number of subgraphs, they can essentially be seen as expectations and estimated through suitable sampling-based approaches.

### B.6 Proof of Proposition 5

We now move to the fully observation setting. Here the algorithm that we analyze performs local maximum likelihood estimation to estimate the parameters of the IC model (see Section 4.1). The specific objective optimized is restated below:

$$
\begin{aligned}
\sum_{i=1}^{m} \mathcal{L}(X^i, Y_1^i; \mathbf{w}) &= \sum_{i=1}^{m} \sum_{u \notin X^i} \left[ \chi_u(Y_1^i) \ln \left( f_u^{\mathbf{w}}(X^i) \right) + (1 - \chi_u(Y_1^i)) \ln \left( 1 - f_u^{\mathbf{w}}(X^i) \right) \right] \\
&= \sum_{i=1}^{m} \sum_{u=1}^{n} \left[ \chi_u(Y_1^i) \ln \left( f_u^{\mathbf{w}}(X^i) \right) + (1 - \chi_u(Y_1^i)) \ln \left( 1 - f_u^{\mathbf{w}}(X^i) \right) \right] \mathbf{1}(u \notin X^i),
\end{aligned}
$$

$$(9)$$

where notice that the likelihood is not evaluated on nodes that are already present in the seed set. Note that we did not have this issue with the partial observation case, as there the global influence function $F_u(X)$, by definition, would evaluate to 1 whenever $X$ contains $u$ (as $u$ is influenced even before the cascade begin; see Eq. (2)). On the other hand, the local influence function $f_u^{\mathbf{w}}(X)$ need not evaluate to 1 when $u \in X$ and hence this case is ignored in the above objective.

Our analysis involves first showing guarantees on the estimated parameters, and transferring them to guarantees on the global IC influence function. Unlike the partial observation case, here we seek to derive optimality guarantees on the parameters themselves, and require stricter assumptions.

**Discussion on Assumption 1** In particular, the following are the assumptions we make:

1. All edges have a minimum influence strength of $\lambda$. Note that the graph can still contain a node that has no influence on its neighbor, by not having an edge between the two nodes.

2. Even when all neighbors of a node are influenced in a time step, there is a small probability $\gamma > 0$ of the node not being influenced in the next step. Thus expect for the case where none of a node's neighbors are present in the seed set, there is always a small probability of the node not being influenced in the first time step.

3. The seed distribution is such that each node is chosen independently with probability $\kappa \in (0, 1)$.

The first and second assumptions ensure that the IC influence function and hence the log-likelihood function is bounded, a property which is crucial to guarantee learnability. The third assumption avoids pathological cases where the support of the seed distribution only covers a subset of nodes (in which case, we will not be able to learn anything about the remaining nodes), or has its entire probability mass concentrated on the full set $V$ (in which case, we again learn nothing about the individual edge probabilities). Indeed our analysis will go through if in place of the second assumption, we just restricted the edge probabilities to be upper bounded by a value below 1, and the third assumption allows for more general distributions with appropriate support. We have retained these slightly stricter assumptions so that analysis is cleaner and easier to follow.

We begin rewriting the local influence functions $f_u^{\mathbf{w}}$ in terms of transformed parameters $\beta_{uv} = -\ln(1 - w_{uv})$: $f_u^{\beta}(Z) = 1 - \exp\left(-\sum_{v \in N(u) \cap Z} \beta_{uv}\right)$, where $\sigma(s) = 1 - \exp(-s)$. Let us use the notation $\boldsymbol{\beta}_u$ to denote the vector of parameters $\beta_{uv}, v \in N(u)$. Also, recall that the prescribed local estimation procedure solves an optimization over all parameters that satisfy Assumption 1. Due to this, in our analysis, we can safely assume that the parameters are bounded in a certain range. In particular, it is clear that for all $(u, v) \in E$, $w_{uv} \geq \lambda$. One can also derive an upper bound from Assumption 1 as follows: for any $(u, v) \in E$, $w_{uv} = 1 - (1 - w_{uv}) \leq 1 - \prod_{v' \in N(u)}(1 - w_{uv'}) \leq 1 - \gamma$. Translating these bounds to the log-transformed space, we conclude that the log-transformed parameters $\boldsymbol{\beta} \in \mathbb{R}_+^r$ satisfy: $-\ln(1 - \lambda) \leq \beta_{uv} \leq -\ln(\gamma)$.

We are now ready to sketch the proof of Proposition 5. Let $\overline{\mathbf{w}}$ be the parameters obtained by local ML estimation. We shall show guarantees on $\overline{\mathbf{w}}$ and translate them to guarantees on $F^{\overline{\mathbf{w}}}$.

- We first establish an $\epsilon$-cover of local IC influence functions $f_u^{\beta}$ or $f_u^{\mathbf{w}}$, and obtain a bound on the covering number of this space.

   **Lemma 10 (Covering number of local IC influence functions).** *Under Assumption 1, the $L_\infty$ covering number of the class of all local IC influence functions $f_u^{\beta}$ for a node $u$ with $n_u$ parameters and radius $\epsilon$ is $O((\ln(1/\gamma)/\epsilon)^{n_u})$.*

   *Proof.* See Section B.7. □

- The covering number result allows us to invoke standard uniform convergence arguments to prove that the log-likelihood of the $\overline{\mathbf{w}}$ can be taken arbitrarily close to the optimal value. But, this does not imply that the estimated parameters are themselves close to the optimal parameters. For this, we show that under Assumption 1, the expected log-likelihood objective is strongly concave in the IC parameters, or equivalently that the negative likelihood is strongly convex, which then implies the desired result.

   **Lemma 11 (Guarantees on parameters obtained by local ML estimation).** *Let $\mathbf{w}^* \in [0, 1]^r$ be the true IC parameters and $\overline{w}_{uv} = 1 - \exp(-\beta_{uv})$ be obtained by local ML estimation. Fix $\epsilon, \delta \in (0, 1)$. Under Assumption 1, if $m = \widetilde{O}(nr(\kappa^2(1-\kappa)^4\lambda^2\gamma^2\epsilon^2)^{-1})$, with probability at least $1 - \delta$ (over draw of the training sample), $\forall (u, v) \in E$, $\|\mathbf{w}^* - \overline{\mathbf{w}}\|_2^2 \leq \epsilon$.*

   *Proof.* See Section B.8. □

- Given that that the global IC influence function is Lipschitz (see Lemma 3), the above guarantees translate to the following sample complexity guarantee on the IC influence function.

  **Lemma 12** (**Translation to global influence function**). *Under the statement of Lemma 11, we have with probability at least* $1 - \delta$, $\left( F_u^{\mathbf{w}^*}(X) - F_u^{\overline{\mathbf{w}}}(X) \right)^2 \leq r\epsilon, \ \forall u \in V, \ X \subseteq V.$

  *Proof.* See Section B.9 $\qquad\square$

Proposition 5 then directly follows from the above sequence of results.

*Proof of Proposition 5.* Let $\mathbf{w}^* \in \mathbb{R}_+^r$ be the parameters of the underlying IC model satisfying Assumption 1. Fix $\epsilon, \delta \in (0, 1)$, and let $\overline{\mathbf{w}}$ be the parameters obtained from the local maximum likelihood estimation. From Lemma 12, when the number of training examples $m = \widetilde{O}(nr(\kappa^2(1 - \kappa)^4 \lambda^2 \gamma^2 \epsilon^2)^{-1})$, we have with probability at least $1 - \delta$ (over draw of training sample), for each node $u \in V$, and seed set $X \subseteq V$:

$$\left( F_u^{\mathbf{w}^*}(X) - F_u^{\overline{\mathbf{w}}}(X) \right)^2 \leq r\epsilon,$$

As in the proof of Theorem 2 (see Eq. (4)), we can show from this that with probability at least $1 - \delta$,

$$\mathrm{err}^{\mathrm{sq}}\left[ F^{\overline{\mathbf{w}}} \right] - \inf_{\mathbf{w} \in \mathbb{R}_+^r} \mathrm{err}^{\mathrm{sq}}\left[ F^{\mathbf{w}} \right] \ \leq \ \frac{1}{n} \sum_{u=1}^n \mathbf{E}_X[r\epsilon] \ \leq \ r\epsilon.$$

Absorbing $r$ on the right hand side into the sample complexity bound, gives us the desired result. $\quad\square$

## B.7 Proof of Lemma 10

*Proof.* The local IC influence function for any $Z \subseteq V$ is $f_u^\beta(Z) = 1 - \exp\left( -\sum_{v \in N(u) \cap Z} \beta_{uv} \right) = \sigma(\sum_{v \in N(u) \cap Z} \beta_{uv})$, which is a linear function composed with link function $\sigma(s) = 1 - \exp(-s)$. It is well-known that the class of all linear functions with $n_u$ parameters in a bounded range $[a, b]$, can be covered with at most $O(((b - a)/\epsilon)^{n_u})$ $L_1$-balls of radius $\epsilon$ [20]. In our case, each $\beta_{uv} \in [-\ln(1 - \lambda), -\ln(\gamma)]$, and the number of $L_1$-balls to cover the space of all linear functions defined by parameters in this range is at most $O((\ln(1/\gamma)/\epsilon)^n)$ as $1 - \lambda < 1$. Let $\mathbf{a}_1, \ldots, \mathbf{a}_R$ be the corresponding centers. Now, since $\sigma$ is 1-Lipschitz on the positive real-line (follows from $\sigma'(s) = \exp(-s) \leq 1$ for all $s \geq 0$), a set of $L_\infty$-balls of radius $\epsilon$ with centers $f_u^{\mathbf{a}_1}, \ldots, f_u^{\mathbf{a}_R}$ would then constitute an $\epsilon$-cover over the class of all local IC influence functions. Thus the $L_\infty$ covering number of this function class for radius $\epsilon$ is at most $O\left( \left( \ln(1/\gamma)/\epsilon \right)^{n_u} \right)$. $\quad\square$

## B.8 Proof of Lemma 11

As with the partial observation setting, the proof makes use of standard covering number based uniform convergence result for empirical risk minimization (or equivalently for log-likelihood maximization) over a real-valued function class [20]. To apply these standard results, we must ensure the local log-likelihood is bounded and Lipschitz. We do this below.

As before, let $n_u = |N(u)|$ and define for any $Z \subseteq V$, $y \in \{0, 1\}$, and $u \in V$, the local log-likelihood $g_u$ for parameters $\beta$ as $g_u(Z, y; \beta) = \left[ y \ln \left( f_u^\beta(Z) \right) + (1 - y) \ln \left( 1 - f_u^\beta(Z) \right) \right] \mathbf{1}(u \notin Z)$; the indicator term automatically ignore cases where $u$ is already present in seed set $Z$. Note that for a cascade $(X, Y)$, the local log-likelihood $\mathcal{L}(X, Y_1; \beta) = \frac{1}{n} \sum_{u=1}^n g_u(X, \chi_u(Y_1); \beta)$. Whenever we refer to a subset $Z \subseteq V$ in the context of a node $u$, *we shall assume that $Z$ contains a neighbor of $u$*; cases where this assumption fails can be easily handled, but have been ignored here to make the proof more accessible. Below, we show that $g_u$ is bounded and Lipschitz for any $u$.

**Lemma 13** (**Boundedness and Lipschitz continuity of log-likelihood function**). *Let $\beta$ be obtained from edge weights that satisfy Assumption 1. Then*

1. $\lambda \leq f_u^\beta(Z) \leq 1 - \gamma$.

2. $|g_u(Z, y; \beta)| \leq \ln(1/\gamma)$.

3. $g_u(Z, y; \beta)$ is $1/\lambda$-Lipschitz in $\beta_u$ w.r.t. the $L_1$ norm.

*Proof.*

1. Starting with the upper bound, we have $f_u^{\boldsymbol{\beta}}(Z) = 1 - \exp\left(-\sum_{v\in N(u)\cap Z}\beta_{uv}\right) \leq 1 - \exp\left(-\sum_{v\in N(u)}\beta_{uv}\right) \leq 1 - \gamma$ (by Assumption 1). For the lower bound, $f_u^{\boldsymbol{\beta}}(Z) = 1 - \exp\left(-\sum_{v\in N(u)\cap Z}\beta_{uv}\right) \geq 1 - \exp\left(-\beta_{uv'}\right) \geq \lambda$, where $u'$ is some neighbor of $u$ in $Z$, which we have assumed exists.

2. Using the above result, $|g_u(Z, y; \boldsymbol{\beta})| \leq |y\ln\left(f_u^{\boldsymbol{\beta}}(Z)\right) + (1-y)\ln\left(1 - f_u^{\boldsymbol{\beta}}(Z)\right)| \leq |y\ln(\lambda) + (1-y)\ln(\gamma)| \leq |\ln(\gamma)| = \ln(1/\gamma)$, where we have used $0 < \gamma \leq \lambda < 0.5$.

3. To show Lipschitzness of $g_u$ w.r.t. $L_1$ norm, we bound the $L_\infty$ norm of its gradient w.r.t. $\boldsymbol{\beta}_u$. Let $\widetilde{\mathbf{Z}} \in \{0,1\}^{n_u}$ be a boolean vector whose entries are $\mathbf{1}(v \in Z)$ for each neighbor $v \in N(u)$. Then $g_u(Z, y; \boldsymbol{\beta}) = \left[y\ln\left(1 - \exp(-\widetilde{\mathbf{Z}}^\top\boldsymbol{\beta}_u)\right) - (1-y)\widetilde{\mathbf{Z}}^\top\boldsymbol{\beta}_u\right]\mathbf{1}(u \notin Z)$ and

$$\nabla_{\boldsymbol{\beta}_u}\left[g_u(Z, y; \boldsymbol{\beta})\right] = \mathbf{1}(u \notin Z)\left[\frac{y\exp(-\widetilde{\mathbf{Z}}^\top\boldsymbol{\beta}_u)}{1 - \exp(-\widetilde{\mathbf{Z}}^\top\boldsymbol{\beta}_u)} - (1-y)\right]\widetilde{\mathbf{Z}}$$

$$= \mathbf{1}(u \notin Z)\left[\frac{y\left(1 - f_u^{\boldsymbol{\beta}}(Z)\right)}{f_u^{\boldsymbol{\beta}}(Z)} - (1-y)\right]\widetilde{\mathbf{Z}}.$$

Then we have,

$$\left\|\nabla_{\boldsymbol{\beta}_u}\left[g_u(Z, y; \boldsymbol{\beta})\right]\right\|_\infty \leq \frac{1 - f_u^{\boldsymbol{\beta}}(Z)}{f_u^{\boldsymbol{\beta}}(Z)}\max_{v\in N(u)}\mathbf{1}(v \in Z) \leq \frac{1}{\lambda},$$

where the numerator is upper bounded by 1, and in the denominator we have used the lower bound on $f_u^{\boldsymbol{\beta}}$ shown in the part 1. Hence $g_u$ is $1/\lambda$-Lipschitz in $\boldsymbol{\beta}_u$ w.r.t. the $L_1$ norm.

$\square$

The above boundedness and Lipschitzness properties of the log-likelihood function will enable us to apply standard covering number arguments to show that the log-likelihood of the estimated parameters can be taken close to the optimal value. This does not however imply that the estimated parameters themselves converge to the optimal parameters; in order to show this, we will need the (negative) likelihood objective to additional be strongly convex. We next show that under Assumption 1, the expected (negative) log-likelihood is strongly convex. In particular, define for any $Z \subseteq V$ and $\eta \in [0,1]$, $\widetilde{g}_u(Z, \eta; \boldsymbol{\beta}) = \left[\eta\ln\left(f_u^{\boldsymbol{\beta}}(Z)\right) + (1-\eta)\ln\left(1 - f_u^{\boldsymbol{\beta}}(Z)\right)\right]\mathbf{1}(u \notin Z)$; again the indicator term automatically ignore cases where $u$ is present in seed set $Z$. Then we have:

**Lemma 14 (Strong convexity of negative expected log-likelihood).** *Let $\mu$ be a distribution over subsets of nodes in $V$ and $\boldsymbol{\beta}^*$ be underlying IC parameters, both satisfying Assumption 1. Let $\eta : 2^V \to [0,1]$ with $\eta(Z) \geq \lambda$. Then $\mathbf{E}_{Z\sim\mu}\left[-\widetilde{g}_u\left(Z, \eta(Z); \boldsymbol{\beta}\right)\right]$ is strongly convex in $\boldsymbol{\beta}_u$ and the strong convexity parameter is at least $\gamma\lambda\kappa(1 - \kappa)^2$.*

*Proof.* As in the previous lemma, we use $\widetilde{\mathbf{Z}} \in \{0,1\}^{n_u}$ to denote a boolean vector whose entries are $\mathbf{1}(v \in Z)$ for each neighbor $v \in N(u)$. To show strong convexity of $-\mathbf{E}_Z\left[\widetilde{g}_u\left(Z, \eta(Z); \boldsymbol{\beta}\right)\right]$, we compute its Hessian w.r.t. $\boldsymbol{\beta}_u$ and show that the Hessian is well-conditioned or that its smallest Eigen value is bounded above zero. The Hessian is given by:

$$\nabla_{\boldsymbol{\beta}_u}^2\left[-\mathbf{E}_Z\left[\widetilde{g}_u\left(Z, \eta(Z); \boldsymbol{\beta}\right)\right]\right] = \nabla_{\boldsymbol{\beta}_u}^2\left[-\mathbf{E}_Z\left[\mathbf{1}(u \notin Z)\left[\eta(Z)\ln\left(f_u^{\boldsymbol{\beta}}(Z)\right)\right.\right.\right.$$

$$\left.\left.\left. + (1 - \eta(Z))\ln\left(1 - f_u^{\boldsymbol{\beta}}(Z)\right)\right]\right]\right]$$

$$= \mathbf{E}_Z\left[\mathbf{1}(u \notin Z)\frac{\eta(Z)\exp(-\widetilde{\mathbf{Z}}^\top\boldsymbol{\beta}_u)}{\left(1 - \exp(-\widetilde{\mathbf{Z}}^\top\boldsymbol{\beta}_u)\right)^2}\widetilde{\mathbf{Z}}\widetilde{\mathbf{Z}}^\top\right]$$

$$= \mathbf{E}_Z\left[\mathbf{1}(u \notin Z)\frac{\eta(Z)\left(1 - f_u^{\boldsymbol{\beta}}(Z)\right)}{f_u^{\boldsymbol{\beta}}(Z)^2}\widetilde{\mathbf{Z}}\widetilde{\mathbf{Z}}^\top\right].$$

The following can then be verified to be the smallest Eigen value of the Hessian. Here $\mathbf{x} \in \mathbb{R}^{n_u}$ and we have used for any $Z \subseteq V$ and $u \in V$, $Z_u = \mathbf{1}(u \in Z)$:

$$
\inf_{\mathbf{x}^\top \mathbf{x} = 1} \mathbf{E}_Z \left[ \frac{\eta(Z)\left(1 - f_u^{\boldsymbol{\beta}}(Z)\right)}{f_u^{\boldsymbol{\beta}}(Z)^2} \left(\widetilde{\mathbf{Z}}^\top \mathbf{x}\right)^2 \mathbf{1}(u \notin Z) \right]
$$

$$
\geq \quad \gamma \inf_{\mathbf{x}^\top \mathbf{x} = 1} \mathbf{E}_Z \left[ \eta(Z)\left(\widetilde{\mathbf{Z}}^\top \mathbf{x}\right)^2 \mathbf{1}(u \notin Z) \right]
$$

$$
= \quad \gamma(1 - \kappa) \inf_{\mathbf{x}^\top \mathbf{x} = 1} \mathbf{E}_Z \left[ \eta(Z)\left(\widetilde{\mathbf{Z}}^\top \mathbf{x}\right)^2 \big| u \notin Z \right]
$$

$$
= \quad \gamma(1 - \kappa) \inf_{\mathbf{x}^\top \mathbf{x} = 1} \mathbf{E}_Z \left[ \eta(Z) \sum_{v \in N(u)} Z_v x_v^2 + \eta(Z) \sum_{v \in N(u)} \sum_{k \in N(u)} Z_u Z_k x_v x_k \, \Big| \, u \notin Z \right]
$$

$$
= \quad \gamma(1 - \kappa) \inf_{\mathbf{x}^\top \mathbf{x} = 1} \left[ \sum_{v \in N(u)} \mathbf{E}_Z \left[ \eta(Z) Z_v \, \big| u \notin Z \right] x_v^2 + \sum_{v \in N(u)} \sum_{k \in N(u)} E_Z \left[ \eta(Z) Z_v Z_k \, \big| u \notin Z \right] x_v x_k \right]
$$

$$
\geq \quad \gamma(1 - \kappa) \inf_{\mathbf{x}^\top \mathbf{x} = 1} \left[ \sum_{v \in N(u)} \lambda \mathbf{P}\left(v \in Z \, \big| u \notin Z\right) x_v^2 + \sum_{v \in N(u)} \sum_{k \in N(u)} \lambda \mathbf{P}\left(v \in Z, k \in Z \, \big| u \notin Z\right) x_v x_k \right]
$$

$$
= \quad \gamma \lambda (1 - \kappa) \inf_{\mathbf{x}^\top \mathbf{x} = 1} \left[ \kappa \sum_{v \in N(u)} x_v^2 + \kappa^2 \sum_{v \in N(u)} \sum_{k \in N(u)} x_v x_k \right]
$$

$$
= \quad \gamma \lambda (1 - \kappa) \inf_{\mathbf{x}^\top \mathbf{x} = 1} \left[ (\kappa - \kappa^2) \sum_{v \in N(u)} x_v^2 + \kappa^2 \left( \sum_{v \in N(u)} x_v \right)^2 \right]
$$

$$
\geq \quad \gamma \lambda (1 - \kappa) \left[ (\kappa - \kappa^2)(1) + 0 \right]
$$

$$
\geq \quad \gamma \lambda \kappa (1 - \kappa)^2,
$$

where the first step follows from $f_u^{\boldsymbol{\beta}}(Z) \leq 1 - \gamma < 1$, the second and sixth step follow from $Z$ being drawn from a distribution that satisfies Assumption 1 (i.e. a distribution where each element in $V$ is chosen independently w.p. $\kappa \in (0, 1)$), and the fifth step follows from $\eta(Z) \geq \lambda$. Thus the given expected log-likelihood is strongly convex in $\boldsymbol{\beta}_u$ under the given assumptions, and the strong convexity parameter is at least $\gamma \lambda \kappa (1 - \kappa)^2$. $\qquad \square$

We now make use of both the above results to prove Lemma 11.

*Proof of Lemma 11.* For the parameters $\overline{\mathbf{w}}$ obtained by from local ML estimation, define log-transformed parameters $\overline{\beta}_{uv} = -\ln(1 - \overline{w}_{uv})$ (these parameters satisfy Assumption 1 due to the way we have framed the optimization problem). Similarly, let $\boldsymbol{\beta}^* \in \mathbb{R}_+^r$ be the transformed version of the underlying model parameters $\mathbf{w}^*$ (satisfying Assumption 1). We shall begin by making use of standard uniform convergence result based on covering numbers [20] and show that the expected log-likelihood of the obtained parameters $\overline{\beta}$ is close to that of the true parameters $\boldsymbol{\beta}^*$; here we will use the covering number result in Lemmas 10 and the boundedness/Lipschitz properties in Lemma 13. We will then exploit the strong convexity of the expected (negative) log-likelihood (shown in Lemma 14) to translate these bounds to guarantees on the parameters themselves.

First, let us write the empirical (local) log-likelihood objective for the first step optimized by the prescribed procedure (shown in Eq. (9)) in terms of $g_u$.

$$
\frac{1}{mn} \sum_{i=1}^{m} \mathcal{L}(X^i, Y_1^i; \boldsymbol{\beta}) = \frac{1}{n} \sum_{u=1}^{n} \underbrace{\frac{1}{m} \sum_{i=1}^{m} g_u(X^i, \chi_u(Y_1^i); \boldsymbol{\beta})}_{\widehat{G}_u(\boldsymbol{\beta}_u)}.
$$

Since each $\widehat{G}_u$ involves a different set of parameters, they can be essentially maximized independently. The maximizer of the above empirical log-likelihood is then simply a concatenation of the maximizers $\overline{\boldsymbol{\beta}}_u \in \mathbb{R}_+^{n_u}$ of each $\widehat{G}_u$. Similarly, one can write down the expected log-likelihood in

terms of $g_u$:

$$\frac{1}{n}\mathbf{E}_{X,Y_1}\big[\mathcal{L}(X,Y_1;\boldsymbol{\beta})\big] \;=\; \frac{1}{n}\sum_{u=1}^{n}\mathbf{E}_{X,Y_1}\big[\underbrace{g_u(X,\chi_u(Y_1);\boldsymbol{\beta})}_{G_u(\boldsymbol{\beta}_u)}\big].$$

Again each $G_u$ can be maximized independently; the optimal parameters $\boldsymbol{\beta}^*$ for the above objective is given by a concatenation of the optimal parameters $\boldsymbol{\beta}^*_u$ for each $G_u$.

Now from the properties stated in Lemma 13, we have that $g_u$ is bounded by $\ln(1/\gamma)$ and is $1/\lambda$-Lipschitz in $\boldsymbol{\beta}$. We then have based on the covering number result in Lemma 10 for the class of local influence functions, followed by an application of a union bound over all nodes in $V$, that when $m = O\bigg(n_u \ln(1/\gamma)^2 \dfrac{\ln(\ln(1/\gamma)/\lambda\epsilon) + \ln(n/\delta)}{\epsilon^2}\bigg)$, with probability at least $1-\delta$ (over draw of training sample), for each $u \in V$

$$|G_u(\overline{\boldsymbol{\beta}}_u) \;-\; \widehat{G}_u(\overline{\boldsymbol{\beta}}_u)| \;\leq\; \epsilon,$$

where $n_u = |N(u)|$. Equivalently, when $m = O\bigg(\ln(1/\gamma)^2 \dfrac{\ln(\ln(1/\gamma)/\lambda\epsilon\sqrt{n_u}) + \ln(n/\delta)}{\epsilon^2}\bigg)$, with probability at least $1-\delta$ (over draw of training sample), for each $u \in V$

$$|G_u(\overline{\boldsymbol{\beta}}_u) \;-\; \widehat{G}_u(\overline{\boldsymbol{\beta}}_u)| \;\leq\; \epsilon\sqrt{n_u},$$

which will further give us using straight-forward algebra (see proof of Theorem 2) that with probability at least $1-\delta$,

$$G_u(\boldsymbol{\beta}^*_u) \;-\; G_u(\overline{\boldsymbol{\beta}}_u) \;\leq\; \epsilon\sqrt{n_u}. \tag{10}$$

Thus, $\boldsymbol{\beta}^*_u$ and $\overline{\boldsymbol{\beta}}$ are close in terms of their likelihood value. The rest of the proof involves using strong convexity of the (negative) expected log-likelihood (in Lemma 14) to show that similar guarantees hold for the parameters themselves. In particular, we shall use the fact that if the negative of a function $h : \mathbb{R}^d \to \mathbb{R}$ is $q$-strongly convex with $\mathbf{z}^* = \operatorname{argmax}_{\mathbf{z} \in \mathbb{R}^d} h(\mathbf{z})$ the following is true for any $\mathbf{z} \in \mathbb{R}^d$: $h(\mathbf{z}^*) - h(\mathbf{z}) \geq \frac{q}{2}\|\mathbf{z} - \mathbf{z}^*\|_2^2$.

Expanding the left-hand side of the above inequality,

$$
\begin{aligned}
G_u(\boldsymbol{\beta}^*_u) - \widehat{G}_u(\overline{\boldsymbol{\beta}}_u) &= \mathbf{E}_{X,Y_1}\big[g_u(X,\chi_u(Y_1);\boldsymbol{\beta}^*) - g_u(X,\chi_u(Y_1);\overline{\boldsymbol{\beta}})\big]\\
&= \mathbf{E}_{X,Y_1}\big[\mathbf{1}(u \notin X)\big[\chi_u(Y_1)\ln(f_u^{\boldsymbol{\beta}^*}(X)) + (1-\chi_u(Y_1))\ln(1-f_u^{\boldsymbol{\beta}^*}(X))\\
&\qquad\qquad - \chi_u(Y_1)\ln(f_u^{\overline{\boldsymbol{\beta}}}(X)) - (1-\chi_u(Y_1))\ln(1-f_u^{\overline{\boldsymbol{\beta}}}(X))\big]\big]\\
&= \mathbf{E}_X\big[\mathbf{1}(u \notin X)\big[f_u^{\boldsymbol{\beta}^*}(X)\ln(f_u^{\boldsymbol{\beta}^*}(X)) + (1-f_u^{\boldsymbol{\beta}^*}(X))\ln(1-f_u^{\boldsymbol{\beta}^*}(X))\\
&\qquad\qquad - f_u^{\boldsymbol{\beta}^*}(X)\ln(f_u^{\overline{\boldsymbol{\beta}}}(X)) - (1-f_u^{\boldsymbol{\beta}^*}(X))\ln(1-f_u^{\overline{\boldsymbol{\beta}}}(X))\big]\big]f\\
&= \mathbf{E}_X\big[\widetilde{g}_u\big(X, f_u^{\boldsymbol{\beta}^*};\boldsymbol{\beta}^*\big) - \widetilde{g}_u\big(X, f_u^{\boldsymbol{\beta}^*};\overline{\boldsymbol{\beta}}\big)\big]\\
&\geq \frac{\lambda\gamma\kappa(1-\kappa)^2}{2}\|\boldsymbol{\beta}^*_u - \overline{\boldsymbol{\beta}}_u\|_2^2,
\end{aligned}
$$

where the second equality follows from $E[\chi_u(Y_1)|X] = f_u^{\boldsymbol{\beta}^*}(X)$, and the fourth step follows from the strong convexity result in Lemma 13 with $\eta = f_u^{\boldsymbol{\beta}^*}$. Substituting this back in Eq. (10), we have with probability at least $1-\delta$ (over draw of the training sample), for each $u$

$$\|\boldsymbol{\beta}^*_u - \overline{\boldsymbol{\beta}}_u\|_2^2 \;\leq\; \frac{2\epsilon\sqrt{n_u}}{\lambda\gamma\kappa(1-\kappa)^2}.$$

In other words, if $m = O\bigg(\ln(1/\gamma)^2 \dfrac{\ln(\ln(1/\gamma)/\lambda\epsilon\sqrt{n}) + \ln(n/\delta)}{\kappa^2(1-\kappa)^4\lambda^2\gamma^2\epsilon^2}\bigg)$, then w.p. at least $1-\delta$,

$$\|\boldsymbol{\beta}^*_u - \overline{\boldsymbol{\beta}}_u\|_2^2 \;\leq\; \epsilon\sqrt{n_u}.$$

Summing this over all $u \in V$,

$$\|\boldsymbol{\beta}^* - \overline{\boldsymbol{\beta}}\|_2^2 \;\leq\; \epsilon\sum_{u=1}^{n}\sqrt{n_u} \;\leq\; \epsilon\sqrt{n\sum_{u=1}^{n}n_u} \;=\; \epsilon\sqrt{nr},$$

where the second last step follows from Jensen's inequality given that the square-root is a concave function. We thus have guarantees on the log-transformed parameters. It is straight-forward to show that the same guarantees also hold in the original parameter space, i.e. w.p. at least $1 - \delta$, $\|\mathbf{w}^* - \overline{\mathbf{w}}\|_2^2 \leq \epsilon\sqrt{nr}$. Absorbing the term $\sqrt{nr}$ into the sample complexity bound completes the proof. $\qquad\square$

### B.9 Proof of Lemma 12

*Proof.* Recall from the Lemma 3 that $F_u^{\overline{\mathbf{w}}}$ is 1-Lipschitz in $\mathbf{w}$ w.r.t. the $\ell_1$ norm. So if $\|w^* - \overline{w}\|_2^2 \leq \epsilon$ for all $(u, v) \in E$, then $\left|F_u^{\mathbf{w}^*}(X) - F_u^{\overline{\mathbf{w}}}(X)\right|^2 \leq \|\mathbf{w}^* - \overline{\mathbf{w}}\|_1^2 \leq r\|\mathbf{w}^* - \overline{\mathbf{w}}\|_2^2 \leq r\epsilon$. This, together with the result in Lemma 11, leads to the desired guarantee on $F_u^{\overline{\mathbf{w}}}$. $\qquad\square$

## Footnotes

[4]Note that even with auxiliary connections with constant weights, the VC-dimension of the given class of neural networks is at most $O((r+n)\ln(r+n))$.