[Reviews · NeurIPS 2015]

Submitted by Assigned_Reviewer_1

This is a "light" review so I am leaving detailed comments to the other reviewers.

AFTER AUTHOR FEEDBACK: I'm upping my score to an 8. I think this will be a nice addition to NIPS.
Summary: This paper is on PAC learning for different influence models (e.g. infection, voting, etc.). The authors do not connect much to the literature on infection modeling, which I found surprising. However, their focus is different. The novelty I feel is in figuring out how to phrase the PAC learning condition and how to pick an appropriate loss function. Some of the discussion is bit fuzzy, such as point 2 on page 5. I admit that I did not understand the connection to neural networks very clearly. However, I think this would be of interest to the NIPS community.

Submitted by Assigned_Reviewer_2

This papers studies the learnability of influence functions in networks from data obtained from different forms of cascade sequences. In particular, a number of seed nodes are marked "influenced" and at every next time step try to influence neighboring nodes, succeeding depending on the "influence function" which must be learned.

The paper presents three classes of influence functions, and for each discusses the learnability.

Two settings are considered: one where only the seed nodes are provided next to the nodes influenced after t steps, and one where all influenced nodes are known for all time steps.

In the first case, as expected, only a sample complexity can be shown, while in the latter case also a computational result can be given.

The derivation seems correct, even though quite a few minor language mistakes appear.

To the best of my knowledge, the presented result is novel.

For the partial information case, the obtained results are clearly of interest.

Details: * line 173: \mathcal{F}_G is a function class rather than a function (and classes are PAC-learnable) ? * after line 208, the text mentions a two-layer neural network, while the explanation gives the impression that there is one layer per time step (hence t layers). * suppl material, line 525: standard the VC -> the standard VC ? * suppl material, line 531: is same -> is the same

Summary: The paper is interesting, well-written, and as far as I could verify correct and novel.

Submitted by Assigned_Reviewer_3

The paper studies PAC learnability of three commonly used network influence models, namely linear threshold model, independent cascade model and voter model. The authors study both the fully observed model where the time steps of the influence events are included, and the partial observation model, where only the set of influenced nodes is given. The authors are able to show PAC learnability of the influence models in all the above cases, and polynomial time optimisation for the fully observed linear threshold and independent cascade models and both fully and partial observed voter model.

Quality: I think the paper is of a good quality, in sofar I have been able to judge. I could not perform detailed check of the proofs in the supplementary material.

Clarity:

The paper is readable after some effort. There are some notation and presentation issues which I list below.

Originality: The idea of providing the PAC analysis is original and welcomed. The proof

require considerable insight to come up with as different setups are required (neural network, Vcdim for LT model, covering numbers, log likelihood for IC, regression for voter model), albeit the proof as such use standard tools.

Significance: The results are significant, because they provide the learning theoretic justification for the widely studies influence function models. It is also foreseeable that the work can lead to new approaches in studying influence in complex networks. Is it for example possible to analyse the context-sensitive case (Su et al. ICML 2014, pp. 442-450) where the influence depends on the type of stimulus, not just the topology and the initial set?

Details:

- line 49-53:

I don't get the purpose of the footnote. In the following you show that the models are PAC-learnable. Here, you wish to imply that they are hard to learn? You do not return to this kind of analysis later to this footnote is not connected to the rest of the paper. I would use the limited space allowed by NIPS by making your main arguments clear(er). - line 61-76: I am not sure this summary is needed, the contributions are clear, perhaps the space would be better used later

- line 69: independent CASCADE model - line 167: You are overloading the subscript of set Y: here Y_t denotes time points, below Y_u denotes a node. This confused me a lot. I would be of great help to the reader if you used some other notation for one of the cases. - line 198: I don't think the subscripts _u should be there on the left-hand side of the equation (or, then I really don't understand) - line 245-246: you ... show ... provably ...

=> you prove - line 251: subscripts _u on the left hand side again... - line 253: why does \bar{Y}_u^i need to defined through the union of time points.

Isn't it so hat the last time point contains all actives nodes so the union should always equal Y_n^i? Or perhaps I miss something. - line 270: I am not completely satisfied with the description. There is not literature reference to the computational hardness and you do not hint how this is established. Why do sigmoidal activations and back-propagation help in resolving the time-points?

- line 416: I would appreciate a literature reference to the standard results

Summary: The paper PAC-learnability of three commonly used influence functions for networks. This fills in a gap in the theoretical understanding of these widely studied models.

Submitted by Assigned_Reviewer_4

Summary: ------- Goal is to learn a model that gives good predictions for how states in a network (i.e the opinions of the nodes) evolve over time. They consider 3 different model classes. Each class seems sufficiently complex to merit consideration in that they allow for a fair amount of heterogeneity in the node behavior (some are more easily influenced than others). They give bounds on the sample complexity and also show in some cases that the learning algorithm has polynomial time complexity. Proofs seem to be based mostly on transforming the problem so that it fits canonical cases already known in the literature.

Quality ---------- I think the results are good, particularly in that they handle both computational and sample complexity.

Clarity -------

The paper is clear on its technical contributions. I wish the practical significance of the problem could have been explained in the paper better, as opposed to just relying on the volume of past citations on the problem. Just a few sentences would have helped.

Originality --------

I'm not well qualified to judge this aspect to be honest, but it seems original to me.

Significance ---------

Technically, the paper is safely above threshold in my opinion. The practical significance is harder to determine, as the authors discuss the problem purely from a technical standpoint. They cite papers showing that cascade prediction is important, but I don't know whether this particular version of the prediction problem solves a practical problem of interest or not. In what application could I expect (and want) to predict which nodes will change their mind on an opinion based on the opinions of certain seed nodes? The models are predictive, not causal, so I would be careful in using this to choose which nodes to "seed" with opinions by advertising. I could imagine a useful model of disease spread being learned this way, although the virulence of the disease would also have to be incorporated.

Summary: Paper is technically good, it shows how to efficiently estimate parameters of fairly complex network process in order to minimize prediction error using training set of past events. A small concern is that practical importance of the problem isn't elucidated here, instead they rely on sheer volume of citations.

Author Feedback
Author rebuttal: We thank the reviewers for their positive and encouraging comments. We will correct all typos pointed out. Below are responses to the the main comments/questions.

Reviewer 1:
Thanks for your comments.

Number of neural network layers: Indeed, the neural network representation of the LT influence function has one layer for each time step - hence in addition to the input layer, there are a total of t+1 layers. The two-layer neural network mentioned after line 208 corresponds to the case when t=1. We will make sure to emphasize that.

Reviewer 3:
Practical significance: The problem of estimating/predicting influence in social networks is critical to applications like viral marketing, where one needs to maximize awareness about a product by selecting a small set of influential users (see e.g. [1]). In regard to correlation vs. causation: you're right, of course, but in regard to causal inference, estimation of counterfactuals plays an important role in potential outcome approaches.

Reviewer 4:
Thank you for the positive comments and we are glad that you found the connection to neural networks interesting.

Reviewer 5:
Literature on infection modelling: The three influence models that we consider are the most well-studied ones. We will add a discussion of additional models studied in the literature.

Reviewer 6:
We will carefully proofread the bibliography.

Reviewer 7:
Thank you for your comments.

Context-sensitive models: This would indeed be an interesting extension to our work.

Subscripts 'u' and 't': We will rework the notation to avoid confusion.

lines 198, 251: Thanks for catching this typo.

\bar{Y} and union over time steps: As mentioned in line 167, each Y_t contains the nodes influenced at time step 't' (i.e., nodes that changed their opinion from 0 to 1 at time step t). The union over time steps in \bar{Y} is therefore necessary to obtain the set of nodes influenced across all time steps.

Use of sigmoid: Here we merely point out that in practice, one can approximate the linear threshold activation functions with sigmoidal functions, as the resulting optimization objective would now become continuous and differentiable and amenable to gradient-based/ back-propagation methods. We will make this clear.

lines 270, 416: we will add references to relevant literature here.